# VISUAL TRANSFORMATION TELLING

## ABSTRACT

In this paper, we propose a new visual reasoning task, called Visual Transformation Telling (VTT). Given a series of states (i.e. images), a machine is required to describe what happened (i.e. transformation) between every two adjacent states. Different from most existing visual reasoning tasks, which focus on state reasoning, VTT concentrates on transformation reasoning. We collect 13,547 samples from two instructional video datasets, i.e. CrossTask and COIN, and extract desired states and transformation descriptions to form a suitable VTT benchmark dataset. After that, we introduce an end-to-end learning model for VTT, named TTNet. TTNet consists of three components to mimic human's cognition process of reasoning transformation. First, an image encoder, e.g. CLIP, reads content from each image, then a context encoder links the image content together, and at last, a transformation decoder autoregressively generates transformation descriptions between every two adjacent images. This basic version of TTNet is difficult to meet the cognitive challenge of VTT, that is to identify abstract transformations from images with small visual differences, and the descriptive challenge, which asks to describe the transformation consistently. In response to these difficulties, we propose three strategies to improve TTNet. Specifically, TTNet leverages difference features to emphasize small visual gaps, masked transformation model to stress context by forcing attention to neighbor transformations, and auxiliary category and topic classification tasks to make transformations consistent by sharing underlying semantics among representations. We adapt some typical methods from visual storytelling and dense video captioning tasks, considering their similarity with VTT. Our experimental results show that TTNet achieves better performance on transformation reasoning. In addition, our empirical analysis demonstrates the soundness of each module in TTNet, and provides some insight into transformation reasoning.

## 1 INTRODUCTION

What will come to your mind when you are given a series of images, e.g. Figure 1? Probably we first notice the content of each image, then we link these images in our mind, and finally conclude a series of events from images, i.e. the whole intermediate process of cooking noodles. In fact, this is a typical reasoning process from states (i.e. single images) to transformation (i.e. changes between images), as described in Piaget's theory of cognitive development (Bovet, 1976; Piaget, 1977). More specifically, children at the preoperational stage (2-7 years old) usually pay their attention mainly to states and ignore the transformations between states, whereas the reverse is true for children at the concrete operational stage (7-12 years old). Interestingly, computer vision is developed through a similar evolution pattern. In the last few decades, image understanding, including image classification, detection, captioning, and question answering, mainly focusing on visual states, has been comprehensively studied and achieved satisfying results.

Now it is time to pay more attention to the visual transformation reasoning tasks. Recently, there have been some preliminary studies (Park et al., 2019; Hong et al., 2021) on transformation. For example, Hong et al. (2021) defines a transformation driven visual reasoning (TVR) task, where both initial and final states are given, and the changes of object properties including color, shape, and position are required to be obtained based on a synthetic dataset. However, the current studies of transformation reasoning remain limited in two aspects. Firstly, the task is defined in an artificial environment that is far from reality. Secondly, the definition of transformation is limited to predefined properties, which cannot be well generalized to unseen or new environments. As a result,

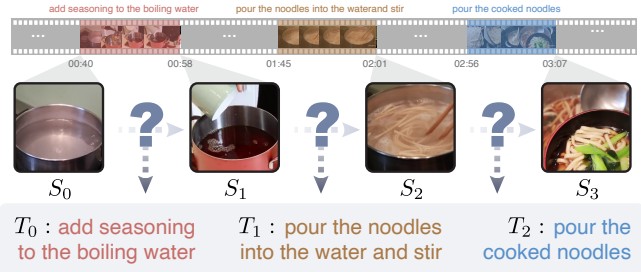

Figure 1: **Visual Transformation Telling (VTT):** given states represented by images (constructed from videos), the goal is to reason and describe transformations between every two adjacent states.

the existing transformation reasoning task cannot meet the requirement of real-world applications. Furthermore, the lack of strong transformation reasoning ability will hinder some more advanced event-level reasoning tasks, such as visual storytelling(Ting-Hao et al., 2016) and procedure planning(Chang et al., 2020), since transformation plays an important role in these tasks.

To tackle these limitations, we propose a new visual transformation telling (VTT) task in this paper. The main motivation is to provide descriptions for real-world transformations. For example, given two images with dry and wet ground respectively, it should be described it rained, which precisely describes a cause-and-effect transformation. Therefore, the formal definition of VTT is to output language sentences to describe the transformation for a given series of states, i.e. images. VTT is different from video description tasks, e.g. dense video captioning Krishna et al. (2017), since the complete process of transformations is shown by videos, which reduces the challenge of reasoning.

To facilitate the study of VTT, we collect 13,547 samples from two instructional video datasets, including CrossTask(Zhukov et al., 2019) and COIN (Tang et al., 2019; 2021). They are originally used for evaluating step localization, action segmentation, and other video analysis tasks. But we found them suitable to be modified to fit VTT, because the transformations are mainly about daily activities, and more importantly, some main steps to accomplish a certain job have been annotated in their data, including temporal boundaries and text descriptions. Therefore, we extract key images from a video as input, and directly use their text labels of the main steps as transformation descriptions. More details can be found in Section 3.2.

When designing an effective VTT model, we face two kinds of challenges. The first one is related to the cognitive challenge, which is to derive abstract transformation from images with small differences, e.g. from the difference between wet and dry ground to rained. The second one is the descriptive challenge, that is, the description of transformations should consider the consistency in a series of images to output a reasonable event. If we only consider the description for a single transformation, i.e. between two images, it is easy to output logical errors in the results.

In order to address these challenges, we propose a difference-sensitive and context-aware model, named TTNet (Transformation Telling Net). TTNet consists of three major components, to mimic the human cognition process of transformation reasoning. To be specific, CLIP (Radford et al., 2021) is utilized as the image encoder to read semantic information from images into image vectors. Then a transformer-based context encoder interacts image vectors together to capture context information. At last, a transformer decoder autoregressively generates descriptions according to context features. However, this basic model is not enough to meet the cognitive and descriptive challenges, so we use three well-designed strategies to improve TTNet. Specifically, the first strategy is to compute difference features on image vectors and fed them into the context encoder as well, to emphasize small visual gaps. Then, masked transformation model is applied to capture the context-aware information, by randomly masking out the inputs of the context encoder like masked language model (Devlin et al., 2019). Finally, in addition to the general text generation loss, the whole network is also supervised under the auxiliary task of category and topic classification, which is to constrain the transformation representations to share underlying semantics, by mimicking human's behavior that forms a global event in mind.

Since the task of VTT is new, there is no ready-made baseline model. Considering the similarity of visual storytelling and dense video captioning to VTT, we modify typical methods including

CST (Gonzalez-Rico & Fuentes-Pineda, 2018), GLACNet (Kim et al., 2019), and Densecap Johnson et al. (2016) in these two applications as our baseline methods. Our experimental results show that TTNet significantly outperforms these methods. Additionally, we conduct comprehensive studies to show the importance of contextual information for VTT and the effectiveness of three strategies, including difference features, masked transformation model, and auxiliary learning.

In conclusion, our major contributions include: 1) the proposal of a new task called visual transformation telling to emphasize the reasoning of transformation in real world applications; 2) the introduction of TTNet which is a difference-sensitive and context-aware model for transformation reasoning. 3) extensive experiments on our collected data from instructional videos, demonstrating the effectiveness of TTNet and providing many insights for understanding the VTT task.

## 2 RELATED WORKS

VTT belongs to the direction of visual reasoning, so we first list some typical visual reasoning tasks and then discuss the relation between VTT and these tasks. CLEVR (Johnson et al., 2017) and GQA (Hudson & Manning, 2019) concentrate on relation and logical reasoning on objects. RAVEN (Zhang et al., 2019) and V-PROM (Teney et al., 2020) care about the induction and reasoning of graphic patterns. VCR (Zellers et al., 2019) and Sherlock (Hessel et al., 2022) test whether machines are able to learn commonsense knowledge to answer daily questions. These tasks mainly concentrate on state-level reasoning. Apart from these tasks, there is a series of works related to dynamic reasoning. Physical reasoning (Bakhtin et al., 2019; Yi et al., 2020; Girdhar & Ramanan, 2020; Baradel et al., 2020; Riochet et al., 2022) evaluates the ability to learn physical rules from data to answer questions or solve puzzles. VisualCOMET (Park et al., 2020) asks to reason beyond the given state to answer what happened before and what will happen next. Visual storytelling (Park et al., 2020) requires completing the missing information between states to describe a story logically. Visual reasoning has a tendency to shift from static scenes to dynamic ones. While state and transformation are both important for reasoning in dynamic scenes, we concentrate on transformation reasoning, between state-only scenarios and more complex composite scenarios.

To the best of our knowledge, there are rare studies on designing specific tasks for visual transformation reasoning. The only work is TVR (Hong et al., 2021). Given the initial and final states, TVR requires to predict a sequence of changes in properties, including size, shape, material, color, and position. However, the synthetic scenario is far from reality and property change is not a common fashion to describe transformations in life. A more natural way is the event-level description. For example, it is more natural to tell it rained when describing what happened between dry and wet ground outside. Visual storytelling (Ting-Hao et al., 2016; Ravi et al., 2021) requires event-level description but transformations are mixed in the description, making it difficult to evaluate transformation only. Visual abductive reasoning (Liang et al., 2022) has a similar core idea to us, which aims to find the most likely explanation for an incomplete set of observations. The difference is they only require machines to reason one single missing transformation from multiple transformations, while our task aims to reason multiple logically related transformations from states. The motivation of procedure planning Chang et al. (2020) is to complete a job given states, while VTT is to explain transformations between states, which has wider scenarios, e.g. explaining the wet ground with rain. Furthermore, the requirement for natural language generation makes VTT have different evaluations and unique challenges such as generalization on language compositions. Walkthrough planning Chang et al. (2020) has a different target which is to predict intermediate states.

Talking about transformation description, there is another topic related, i.e. visual description. Here we review some typical visual description tasks and discuss their differences. Tasks that describe a single image include image captioning (Farhadi et al., 2010; Kulkarni et al., 2011), dense image captioning (Johnson et al., 2016), and image paragraphing (Krause et al., 2017). The difference lies in the level of detail. Similarly, tasks for videos include video description (Venugopalan et al., 2015), video paragraph description (Yu et al., 2016), grounded video description (Zhou et al., 2019), and dense video captioning (Krishna et al., 2017). Different from image captioning tasks that focus only on a single state, video description tasks start to describe events. For example, dense video captioning asks to predict temporal boundaries and descriptions of key events in a video. However, they provide the full process of transformation throughout videos, reducing the need for reasoning.

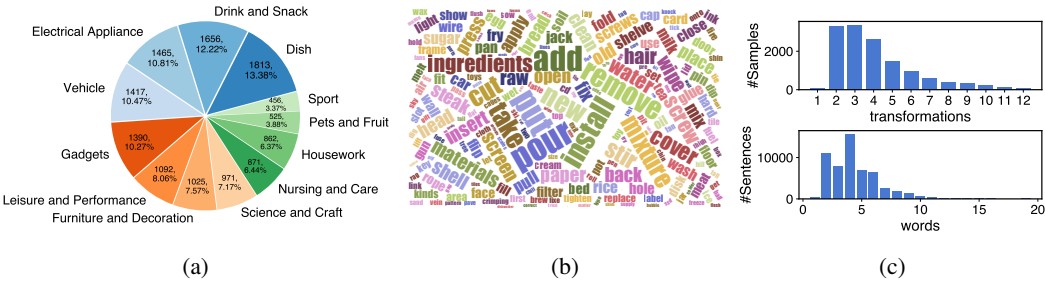

Figure 2: Different distributions of VTT samples. (a) Category. (b) Words. (c) Transformation length (top), and sentence length (bottom).

## 3 VISUAL TRANSFORMATION TELLING

### 3.1 TASK DEFINITION

Visual transformation telling aims to test the ability of machines to reason and describe transformations from a sequence of visual states, i.e. images. Formally, $N + 1$ images $S = \{s_n\}_{n=1}^{N+1}$ are given, which are logically related and semantically different. Logically related means these images are associated with a certain event, e.g. completing a job, while semantically difference is to expect some substantial changes that are meaningful to people, i.e. transformation. The target is then to reason $N$ transformations $T = \{t_n\}_{n=1}^{N}$ between every two adjacent images and describe them with natural languages, so that $s_1 \rightarrow t_1 \rightarrow s_2 \rightarrow \cdots \rightarrow t_n \rightarrow s_{n+1}$ is logically sound.

### 3.2 VTT DATASET

To construct a meaningful dataset for VTT, we require the data to cover a large scope of real world transformations. Therefore, we choose instructional videos as our basic library, because they contain many daily life activities. Specifically, we choose two typical instructional video datasets, i.e. CrossTask (Zhukov et al., 2019) and COIN (Tang et al., 2019; 2021), and construct our data. Figure 1 illustrates an instruction video from COIN for cooking noodles and how we transform their annotation into VTT dataset. We can see that the video is segmented into multiple main steps, and each step is annotated with precise temporal boundaries and text labels. We directly use their text labels as transformation descriptions and extract states based on temporal boundaries. Specifically, for the first transformation, the first frame of the corresponding step segment becomes its start state and the last frame becomes its end state. For the remaining transformations, the end state is extracted in the same way, while the start state shares the end state of the last transformation. We check the quality of states and find transformations can be reasoned out in almost all samples. In this way, we collected 13,547 samples as well as 55,482 transformation descriptions from CrossTask and COIN, forming our new data for VTT. Figure 2 shows the distribution of the sample category, keyword, transformation length, and sentence length. From the category distribution and the word cloud, we can see that VTT data covers lots of daily activities, like dish, drink and snack, electrical application, vehicle, gadgets, leisure and performance, etc. Furthermore, the distribution of transformation length shows its diversity while most of samples contains about 2-5 transformations. Sentence length is around 2-6 on average which means short descriptions make up the majority.

## 4 METHOD

**Problem Formulation.** Our main idea for solving visual transformation telling is to find a parameterized model $f_\theta$ to estimate the conditional probability $p(T|S; \theta) = p(\{t_j\}_{j=1}^{N} | \{s_i\}_{i=1}^{N+1}; \theta)$, where $s_i \in \mathbb{R}^{C \times W \times H}$ is a state represented as an image and $t_j = \{x_{j,l}\}_{l=1}^{L}$ is a sentence of length $L$. The conditional probability can also be written as auto-regressively generating $N$ sentences:

$$p(T|S; \theta) = \prod_{j}^{N} \prod_{l}^{L} p(x_{j,l}|x_{j,<l}, \{s_i\}_{i=1}^{N+1}; \theta) \tag{1}$$

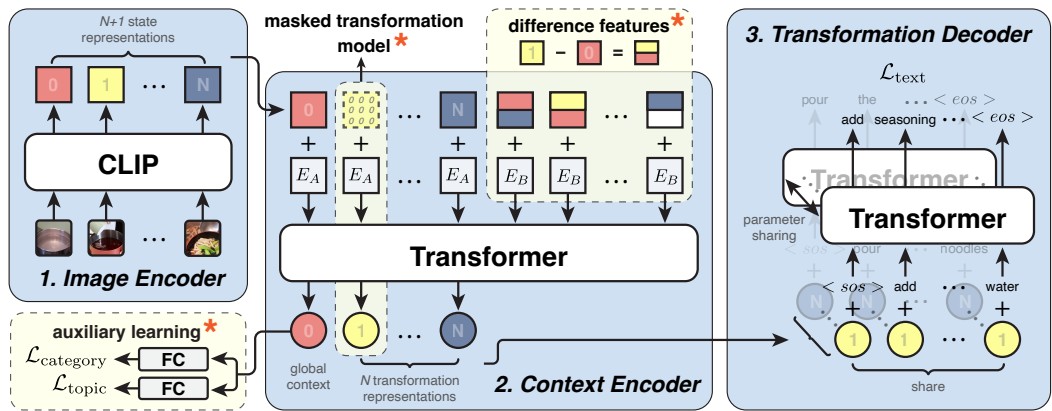

Figure 3: **The architecture of TTNet.** Images are first encoded into state representations in the image encoder, then transformed into transformation representations in the context encoder, and finally decoded into text by the transformation decoder. To be more difference-sensitive and context-aware, three strategies are considered to enhance TTNet (marked with *). Difference features computed according to state representations are used as extra features, state representations and difference features are randomly masked as zero during training, and auxiliary tasks are used for supervision.

**Overview of TTNet.** Our TTNet is designed to mimic human's cognitive process of transformation. The first step is independent recognition, which means that people may understand each image independently. Therefore, we introduce an **image encoder** $f_{\text{image}}$ to represent each image to a vector and obtain a set of image vectors $V = \{v_i\}_{i=1}^{N+1} = \{f_{\text{image}}(s_i)\}_{i=1}^{N+1}$. After that, humans will associate these images together, and form an understanding of all images guided by a global event. To reflect this process, we introduce a **context encoder**, e.g. a bi-directional RNN or a transformer encoder, denoted as $f_{\text{context}}$, to obtain context-aware image representations $C = \{c_i\}_{i=1}^{N+1} = \{f_{\text{context}}(i, V)\}_{i=1}^{N+1}$ by considering contextual information. The final step is to describe these transformations based on previous understanding. In TTNet, we feed the last $N$ context-aware image representations to a **transformation decoder** $f_{\text{caption}}$, implemented with an RNN or a transformer decoder, to generate each transformation description $T = \{t_i\}_{i=1}^{N} = \{f_{\text{caption}}(c_{i+1})\}_{i=1}^{N}$ separately and auto-regressively. We empirically found adding the transformation representation to the word embedding in each step is better than using it as the start token.

The model is then trained with ground truth transformations $T^* = \{t_i^*\}_{i=1}^{N}$ by minimizing the following negative log-likelihood loss, where $t_i^* = \{x_{i,l}^*\}_{l=1}^{L}$ is the ground truth description of the $i_{\text{th}}$ transformation.

$$\mathcal{L}_{\text{text}} = -\sum_{i=1}^{N}\sum_{l=1}^{L} \log p(x_{i,l}^* | x_{i,<l}^*) \tag{2}$$

In order to tackle the two unique challenges of VTT, i.e. cognitive challenge and descriptive challenge, we propose three specific strategies to enhance the above TTNet, including difference sensitive encoding, masked transformation model, and auxiliary learning. To distinguish more clearly, we called the model that does not use these three strategies TTNet$_{\text{base}}$.

## 4.1 DIFFERENCE SENSITIVE ENCODING

In visual transformation telling, the differences between two adjacent states are usually very small. Imagine the scene of cooking noodles, the whole picture does not change much before and after the noodles are added to the pot. This characteristic requires the model not only to understand the content of each image, but also to focus on differences between images to facilitate the understanding of transformations. For this purpose, we first utilize CLIP (Radford et al., 2021) as our image encoder, due to its strong semantic representation ability trained on large scale unsupervised data. We also introduce difference features, by subtracting the current state and the previous state representations $\Delta V = \{v_i - v_{i-1}\}_{i=1}^{N+1}$, where $v_0 = v_{N+1}$, to emphasize the subtle difference. The above two kinds

of representations are concatenated and fed to the context encoder. Furthermore, a type embedding is added to distinguish these two kinds of features.

Since transformations are not independent, we may meet the logical consistency problem in the transformation description process, named the descriptive challenge of VTT. For example, the logic of descriptions does not make sense as shown in Figure 4. TTNet$_{\text{base}}$ recognizes oranges as eggs, which is logically unreasonable with the two transformations before and after. In TTNet, we introduce masked transformation model in the context encoder and auxiliary learning in the loss function to alleviate this problem.

### 4.2    MASKED TRANSFORMATION MODEL

Masked transformation model (MTM) is inspired by masked language model (Devlin et al., 2019). The intuition behind this is that one transformation can be reasoned from nearby transformations. For example, if you are told the previous transformation is washing the watermelon and the next is putting the watermelon into a planet, it is obvious that the intermediate transformation should be related to the watermelon. Following this intuition, 15% of the input features, including

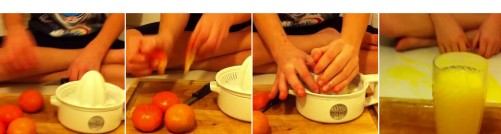

1. Cut both ends and remove fruit seeds.
2. Pour the egg into the bowl.
3. Pour the orange juice into the cup.

Figure 4: TTNet$_{\text{base}}$ failes to effectively use contextual information and mistakenly identifies the orange as an egg.

state representations and difference features, are randomly masked during training. Furthermore, we empirically found that, for each sample, using this strategy with half the probability works better.

### 4.3    AUXILIARY LEARNING

Human usually tries to guess the category or topic before describing transformations, e.g. cooking noodles. Therefore, this category or topic information may help guide description generation. Inspired by this, we propose to leverage auxiliary tasks, i.e. category and topic classification, to supervise the training process. Specifically, we introduce two additional cross entropy losses $\mathcal{L}_{\text{category}}$ and $\mathcal{L}_{\text{topic}}$ on the global context vector. We expect to make the learned transformation representations share the underlying category and topic information to enhance the learning of consistent representations. So the final training loss becomes a combination of $\mathcal{L}_{\text{text}}$, $\mathcal{L}_{\text{category}}$, and $\mathcal{L}_{\text{topic}}$:

$$\mathcal{L} = \mathcal{L}_{\text{text}} + \alpha \mathcal{L}_{\text{category}} + \beta \mathcal{L}_{\text{topic}} \tag{3}$$

where $\alpha$ and $\beta$ are two adjustment factors.

## 5    EXPERIMENTS

In this section, we first introduce our empirical setups including baseline methods and evaluation metrics. Then we demonstrate the main empirical results on the collected VTT dataset, including both quantitative and qualitative results. After that, we show extensive ablation studies on different strategies used in TTNet.

### 5.1    EMPIRICAL SETUPS

**Baseline Models.** Visual storytelling and dense video captioning are the two most similar tasks to VTT. Visual storytelling requests to generate $N$ descriptions from $N$ images. We select two classic methods from the winners of visual storytelling challenge Mitchell et al. (2018), including CST Gonzalez-Rico & Fuentes-Pineda (2018), and GLACNet Kim et al. (2019) for comparison. CST contextualizes image features by LSTM and then generates descriptions with separate LSTMs for each image. GLACNet mixtures global LSTM features and local image features into context features and then generates descriptions with a shared LSTM decoder. When generating transformation descriptions, only the last $N$ context features are used. Dense video captioning has a similar target to describe a series of events. The difference is the input is a video and it additionally requires to predict temporal boundaries for events. We choose DenseCap Johnson et al. (2016) for adaptation which proposed in the paper that introduces dense video captioning. DenseCap integrates the

Table 1: Performance on the test set of VTT dataset. B@4/M/R/C/S/BS/Fl/Re/LS are short for BLEU@4 / METEOR / ROUGE-L / CIDEr / SPICE / BERT-Score / Fluency / Relevance / Logical Soundness. * indicates to use CLIP image encoder for a fair comparison. † indicates TTNet significant ($p < 0.05$) outperforms the corresponding model on this human evaluation metric.

| Model | B@4 | M | R | C | S | BS | Fl | Re | LS |
|---|---|---|---|---|---|---|---|---|---|
| CST | 10.09 | 11.39 | 25.98 | 43.22 | 9.28 | 16.30 | - | - | - |
| CST* | 13.96 | 19.21 | 38.11 | 84.60 | 21.85 | 25.66 | 2.04† | 3.16† | 2.96† |
| GLACNet | 42.77 | 45.26 | 52.98 | 381.48 | 45.33 | 60.12 | - | - | - |
| GLACNet* | 55.24 | 59.48 | 66.25 | 508.18 | 60.21 | 71.13 | 4.75 | 3.82† | 3.78† |
| DenseCap* | 48.25 | 52.00 | 59.79 | 439.68 | 53.73 | 66.30 | 4.74 | 3.67† | 3.59† |
| TTNet$_{Base}$ | 55.68 | 60.47 | 67.05 | 515.12 | 61.45 | 72.22 | **4.79** | 4.04 | 3.95† |
| TTNet | **61.22** | **66.31** | **71.84** | **570.63** | **66.20** | **76.25** | 4.78 | **4.10** | **4.11** |

past and future information into image features to capture the context information. There are many advanced methods for dense video captioning but highly rely on fine-grained video features, which are not suitable for our task. All three methods are implemented as closely as possible according to the original paper and provide a fair comparison by using the same image encoder with TTNet. We describe the implementation details of TTNet as well as baseline models in Section C.

**Evaluation Metrics.** Following previous works on visual descriptions (Ting-Hao et al., 2016; Krishna et al., 2017; Liang et al., 2022), automated metrics including BLEU@4 (Papineni et al., 2002), CIDEr (Vedantam et al., 2015), METEOR (Banerjee & Lavie, 2005), ROUGE-L (Lin & Hovy, 2002), SPICE (Anderson et al., 2016), and BERT-Score (Zhang et al., 2020) are selected as automatic metrics. Furthermore, we asked 25 human annotators to assess the quality of transformation descriptions using a Likert scale ranging from 1 to 5, for following criteria: fluency, measuring how well-written the transformation is; relevance, how relevant the transformations toward the image states; logical soundness, how well the overall logic conforms to common sense.

## 5.2 RESULTS ON VTT DATASET

**Quantitative Results.** Table 1 summarizes the results of 7 models on the VTT dataset, including TTNet, TTNet$_{base}$, CST and its CLIP version, GLACNet and its CLIP version, and CLIP version DenseCap. From the results, TTNet surpasses other models on most metrics with a large margin, e.g. CIDEr is 11% higher than the second best model, i.e. TTNet$_{base}$. This large improvement comes from the three strategies we proposed, which are the only differences between TTNet and TTNet$_{base}$. Further comparing human metrics between them, the main strength of TTNet is the much stronger overall logic of the generated descriptions, while the relevance is only slightly better and the fluency is about the same. Secrefsec:ablation further shows the advantages of TTNet with detailed ablation studies. It is not difficult to find that the performance gap between CST*, GLACNet*, and Densecap* is also very large. While they all use CLIP, the difference lies in the way of context decoding and text generation. GLACNet* outperforms DenseCap* mainly because LSTM captures more information than past and future attention features according to the higher scores of relevance and logical soundness. The gap between GLACNet* and CST* is caused by the way of text generation. GLACNet uses word embeddings and context features as inputs in each LSTM step, while CST only uses the context as the initial state of LSTM. In our empirical studies, this little difference improves the fluency a lot, and it is the reason that TTNet chooses to add context embedding to word embedding as the inputs of the transformation decoder rather than using the context feature as the start token. The underlying design philosophy between TTNet$_{base}$ and GLACNet* is similar, therefore, the performance is close. However, TTNet$_{base}$ converges faster than GLACNet* during training because the transformer captures the context information more efficiently than LSTM.

**Qualitative Results.** We show two examples from the VTT test data in Figure 5 about sowing and pasting a car sticker. From these two examples, we can first realize that the gap between the states is really small. For example, in the sticker case, only a small area of the sticker is changed, making it difficult to reason a certain transformation without considering the overall pasting process. We can see that when the states are confusing, e.g. DenseCap and GLACNet identify the wrong entity in the sow case, TTNet is able to reason the correct transformations from the differences and the

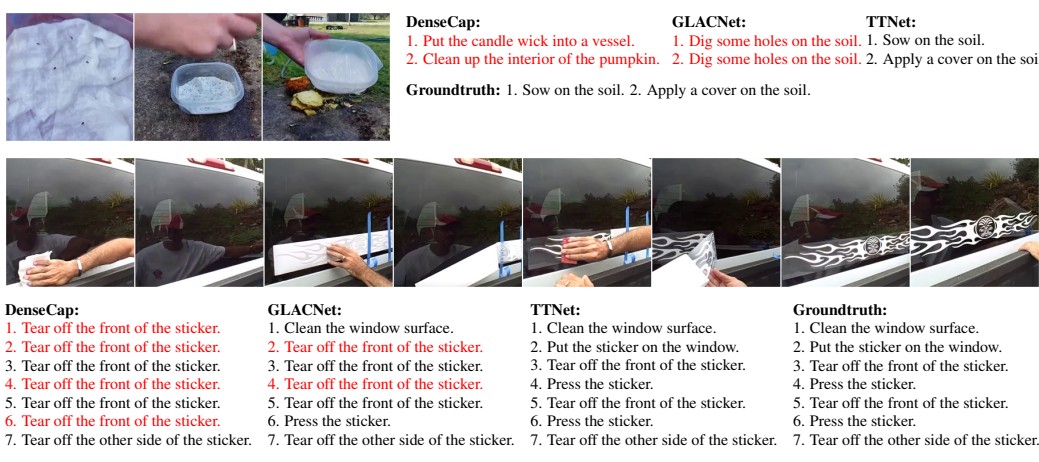

**DenseCap:**
1. Put the candle wick into a vessel.
2. Clean up the interior of the pumpkin.

**GLACNet:**
1. Dig some holes on the soil.
2. Dig some holes on the soil.

**TTNet:**
1. Sow on the soil.
2. Apply a cover on the soil.

**Groundtruth:** 1. Sow on the soil. 2. Apply a cover on the soil.

**DenseCap:**
1. Tear off the front of the sticker.
2. Tear off the front of the sticker.
3. Tear off the front of the sticker.
4. Tear off the front of the sticker.
5. Tear off the front of the sticker.
6. Tear off the front of the sticker.
7. Tear off the other side of the sticker.

**GLACNet:**
1. Clean the window surface.
2. Tear off the front of the sticker.
3. Tear off the front of the sticker.
4. Tear off the front of the sticker.
5. Tear off the front of the sticker.
6. Press the sticker.
7. Tear off the other side of the sticker.

**TTNet:**
1. Clean the window surface.
2. Put the sticker on the window.
3. Tear off the front of the sticker.
4. Press the sticker.
5. Tear off the front of the sticker.
6. Press the sticker.
7. Tear off the other side of the sticker.

**Groundtruth:**
1. Clean the window surface.
2. Put the sticker on the window.
3. Tear off the front of the sticker.
4. Press the sticker.
5. Tear off the front of the sticker.
6. Press the sticker.
7. Tear off the other side of the sticker.

Figure 5: Qualitative comparison on the VTT test data. Above: sow. Below: paste car sticker.

Table 2: CIDEr of independent transformation prediction.

| Model | Original | Indep. |
|---|---|---|
| CST* | 84.90 | 49.80 |
| DenseCap* | 439.53 | 295.75 |
| GLACNet* | 508.19 | 268.49 |
| TTNet w/o diff | 527.62 | **422.04** |
| TTNet | **570.63** | 349.96 |
| TTNet (retrain) | - | 459.84 |

Table 3: Ablation studies on the effect of key components.

| Diff. | MTM | Aux. | C | BS |
|---|---|---|---|---|
| | | | 515.28 | 72.22 |
| $\sqrt{}$ | | | 556.85 | 75.00 |
| | $\sqrt{}$ | | 520.04 | 72.72 |
| | | $\sqrt{}$ | 521.93 | 72.97 |
| $\sqrt{}$ | $\sqrt{}$ | | 562.25 | 75.62 |
| $\sqrt{}$ | | $\sqrt{}$ | 562.83 | 75.72 |
| | $\sqrt{}$ | $\sqrt{}$ | 527.62 | 73.54 |
| $\sqrt{}$ | $\sqrt{}$ | $\sqrt{}$ | **570.63** | **76.25** |

Table 4: Analysis of difference features and auxiliary tasks.

| State | Difference | CIDEr |
|---|---|---|
| $\sqrt{}$ | - | 527.62 |
| $\sqrt{}$ | early | 559.78 |
| $\sqrt{}$ | late | **570.63** |

| Category | Topic | CIDEr |
|---|---|---|
| $\sqrt{}$ | | 549.44 |
| | $\sqrt{}$ | 562.96 |
| $\sqrt{}$ | $\sqrt{}$ | **570.63** |

context. Furthermore, when the difference between states becomes rather small and the transformation length becomes large, TTNet is still able to judge subtle differences between transformations. In contrast, GLACNet indeed understands the topic of pasting the car sticker but fails to distinguish some transformations. In conclusion, TTNet is able to reason transformations from confusing states and distinguish subtle differences between transformations, making it excel other methods.

## 5.3 ABLATION STUDIES

In Section 4, we introduce three strategies to improve TTNet, including difference sensitive encoding, masked transformation model, and auxiliary learning. In this section, we discuss the effectiveness of these three strategies. But before that, we first need to answer the question that whether the context information is crucial for VTT, since all three strategies act on the context encoder to enhance the ability to capture context information. If the answer is yes, then it comes to answer how these strategies work and whether there exist alternative choices, e.g. other types of difference features. Experimental analyses are organized into five following topics according to this logic.

**Importance Analysis of Context.** To answer the question of whether the context is really important for reasoning transformations. We design to let models predict each transformation independently, i.e. only from two states before and after. If transformations can be reasoned without considering the context, model performance should remain roughly the same. However, from Table 2, the CIDEr score of all five models drops sharply from the original setting to the independent setting, showing that the context is clearly very important. Without context, reasoning transformations become rather difficult, and retraining the model with independent data does not help either.

**Effectiveness of Three Strategies.** Next, we move on to analyze the effectiveness of the three strategies and their combinations. The first row in Table 3 shows the result of TTNet$_{\text{Base}}$ and the next

three rows show the results of using each strategy independently on the base model. Among them, the improvement of using difference feature is the most significant, indicating the difference is also crucial for resolving transformation reasoning. The next four rows show the results of combining these strategies and the conclusion is combining all three strategies leads to the best result. The next three topics will go through all these strategies one by one in detail, to see how these strategies work.

**Analysis of Difference Sensitive Encoding.** We just show difference feature is the most significant strategy for TTNet. However, it is not clear how difference features help the TTNet model and if there are alternative choices for difference features, e.g. differences of raw images. To answer the first question, we need to go back to Table 2, which contains an interesting result that TTNet without using difference features overtakes the full model in the independent setting. This phenomenon suggests that difference features help to capture contextual information. Contextual information is more important for the original setting, and the model tries to capture it by attention more to the difference features. However, this does not prevail in the independent setting since contextual information is less effective and the model should attention more to the image features. This is why retraining the full model with the independent data works, because the focus of attention is adjusted during retraining. The second question is about the alternative type of difference features. We compare early and late differences. The early difference is pixel-level difference computed on raw images before input to the image encoder, while the late difference is used by TTNet and computed on encoded image vectors to become the semantic difference. In TVR (Hong et al., 2021), early difference is more effective while Table 4 shows the opposite result. This is because TVR requires to predict property changes on synthetic data, which relies more on pixel differences. In contrast, VTT requires event-level descriptions, with more emphasis on semantic distinctions.

**Analysis of MTM.** We expect MTM to guide the model to reason transformation from nearby transformations. In order to validate this ability, we design to let models predict transformations with incomplete states, e.g mask one state of three. Specifically, we test models under two special settings. In the first setting, we randomly mask one state for all test samples. In the second setting, we give even fewer states on average by only providing start and end states for each sample. The results are shown in Figure 6. We can see that when there is less and less information, the performance of all models decreases. However, TTNet has the slowest decline in perfor-

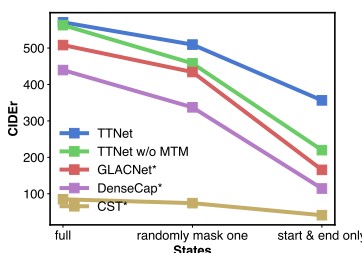

Figure 6: Effect of missing states.

mance, showing its robustness to missing states. By further comparing the results between TTNet and TTNet w/o MTM, we can conclude this robustness is contributed by the MTM strategy.

**Analysis of Auxiliary Tasks.** Finally, we analyze the effects of different auxiliary tasks and report the results in Table 4. From the table, topic classification is more effective than category classification, since topics are more granular than categories. Supervision with two classification tasks simultaneously improves the overall performance, e.g. $562.25 \rightarrow 570.63$ in terms of CIDEr.

## 6 CONCLUSION

This paper introduces a new visual reasoning task to focus on transformation reasoning, i.e. changes between every two states, named visual transformation telling. Given a series of images as states, the description of each transformation is required to represent what happened between every two adjacent images. In this way, the task could be used to test the machines' ability of transformation reasoning, which is an important cognitive skill for humans, as described in Piaget's theory. To the best of our knowledge, this is the first real world application for transformation reasoning by defining transformation descriptions as output. To facilitate the study on VTT, we build benchmark data based on 13,547 samples from two instructional video datasets, i.e. CrossTask and COIN. After that, we design a model named TTNet, by applying three well-designed strategies into a basic human-inspired transformation telling model to make it difference-sensitive and context-aware. From the experiments, we find that the proposed strategies help VTT generate consistent transformation descriptions, and thus obtain better results in terms of natural language generation metrics. The empirical studies provide valuable insights for understanding VTT and the proposed model and may help to design more complicated transformation reasoning tasks or models in the future.

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

## A VTT Dataset Construction

In order to provide a deeper understanding of the VTT dataset, we describe how we construct VTT in detail. The whole process consists of four major parts, i.e. combine and complete annotations, extract state images, and split the dataset. Additionally, we mention some annotation information from CrossTask and COIN, which is important for understanding our decisions when building VTT, such as how to choose the start and end states.

**Combine and complete annotations.** Both CrossTask and COIN provide the annotations of step labels and corresponding segments. Step labels are predefined before they started annotating videos. The difference is that COIN employed experts to define steps while CrossTask derived them from WikiHow, a website teaching how to do many things. With predefined steps, they find annotators to label the step categories and the corresponding segments for each video. We collect and organize these annotations in a uniform format to become the basis of the VTT dataset. Apart from that, category and topic labels are used for auxiliary learning. Both CrossTask and COIN provide topic information, which is the task to solve. However, only COIN provides categories as the domain information and they are missing in the CrossTask. We manually classify all topics from CrossTask into existing categories. The full list of 12 categories and 198 topics is shown in Table 5.

Table 5: The Categories and topics in VTT dataset. Topics marked with * are from CrossTask and others belong to COIN.

| Category | Topics |
| --- | --- |
| Nursing and Care (14) | Wash Dog, Use Earplugs, Use Neti Pot, Put On Hair Extensions, Use Epinephrine Auto-injector, Perform CPR, Wear Contact Lenses, Remove Blackheads With Glue, Give An Intramuscular Injection, Shave Beard, Wash Hair, Bandage Dog Paw, Draw Blood, Bandage Head |
| Pets and Fruit (7) | Plant Tree, Transplant, Graft, Cut Grape Fruit, Cut Mango, Cut Cantaloupe, Sow |
| Furniture and Decoration (15) | Install Shower Head, Install Ceramic Tile, Install Air Conditioner, Install Curtain, Lubricate A Lock, Replace Door Knob, Install Wood Flooring, Install Closestool, Assemble Cabinet, Assemble Sofa, Replace Faucet, Replace Toilet Seat, Assemble Bed, Build Simple Floating Shelves*, Assemble Office Chair |
| Leisure and Performance (17) | Make Paper Wind Mill, Perform Vanishing Glass Trick, Raise Flag, Play Frisbee With A Dog, Make Chinese Lantern, Carve Pumpkin, Change Guitar Strings, Perform Paper To Money Trick, Pitch A Tent, Open Champagne Bottle, Blow Sugar, Make Paper Easter Baskets, Cut And Restore Rope Trick, Do Lino Printing, Replace Drumhead, Prepare Sumi Ink, Prepare Canvas |
| Dish (23) | Make Kimchi Fried Rice*, Cook Omelet, Make Sandwich, Grill Steak*, Clean Fish, Use Toaster, Clean Shrimp, Make Burger, Make French Toast*, Wrap Zongzi, Make French Strawberry Cake*, Make Pickles, Boil Noodles, Make Bread and Butter Pickles*, Make Kerala Fish Curry*, Make Lamb Kebab, Make French Fries, Use Rice Cooker To Cook Rice, Make Pizza, Make Youtiao, Make Salmon, Smash Garlic, Make Pancakes* |
| Electrical Appliance (20) | Replace Graphics Card, Replace Light Socket, Replace Electrical Outlet, Replace Memory Chip, Use Soy Milk Maker, Change Toner Cartridge, Replace Laptop Screen, Replace Refrigerator Water Filter, Use Vending Machine, Replace Filter For Air Purifier, Replace Hard Disk, Replace Blade Of A Saw, Refill Cartridge, Clean Laptop Keyboard, Arc Weld, Install Ceiling Fan, Replace A Bulb, Paste Screen Protector On Pad, Assemble Desktop PC, Use Sewing Machine |
| Science and Craft (15) | Prepare Standard Solution, Make Flower Press, Use Volumetric Pipette, Hang Wallpaper, Make Candle, Make Soap, Use Triple Beam Balance, Make Flower Crown, Use Volumetric Flask, Paste Car Sticker, Make Slime With Glue, Make Paper Dice, Wrap Gift Box, Set Up A Hamster Cage, Use Analytical Balance |
| Drink and Snack (20) | Make Meringue*, Make Salad, Make Lemonade*, Make Taco Salad*, Make Tea, Make Chocolate, Make a Latte*, Make Homemade Ice Cream, Make Jello Shots*, Make Coffee, Make Cocktail, Make Cookie, Make Irish Coffee*, Roast Chestnut, Make Banana Ice Cream*, Make Orange Juice, Make Matcha Tea, Make Sugar Coated Haws, Make Strawberry Smoothie, Make Hummus |
| Vehicle (21) | Change Bike Chain, Replace Car Fuse, Replace Rearview Mirror Glass, Tie Boat To Dock, Pump Up Bicycle Tire, Change Car Tire, Use Jack, Remove Scratches From Windshield, Jack Up a Car*, Change Bike Tires, Install License Plate Frame, Fuel Car, Replace A Wiper Head, Install Bicycle Rack, Replace Tyre Valve Stem, Change a Tire*, Patch Bike Inner Tube, Polish Car, Replace Car Window, Add Oil to Your Car*, Park Parallel |
| Housework (15) | Put On Quilt Cover, Clean Bathtub, Wash Dish, Clean Leather Seat, Pack Sleeping Bag, Clean Wooden Floor, Clean Toilet, Iron Clothes, Drill Hole, Remove Crayon From Walls, Clean Hamster Cage, Make Bed, Unclog Sink With Baking Soda, Clean Rusty Pot, Clean Cement Floor |
| Sport (10) | Practise Karate, Wear Shin Guards, Practise Triple Jump, Throw Hammer, Play Curling, Practise Skiing Aerials, Practise Pole Vault, Attend N B A Skills Challenge, Glue Ping Pong Rubber, Practise Weight Lift |
| Gadgets (21) | Open A Lock With Paperclips, Replace Mobile Screen Protector, Load Grease Gun, Change Mobile Phone Battery, Replace Sewing Machine Needle, Change Battery Of Watch, Replace SIM Card, Resize Watch Band, Replace CD Drive With SSD, Refill Mechanical Pencils, Make Wireless Earbuds, Refill Fountain Pen, Refill A Lighter, Rewrap Battery, Replace Battery On Key To Car, Fix Laptop Screen Scratches, Operate Fire Extinguisher, Replace Battery On TV Control, Use Tapping Gun, Refill A Stapler, Make RJ45 Cable |

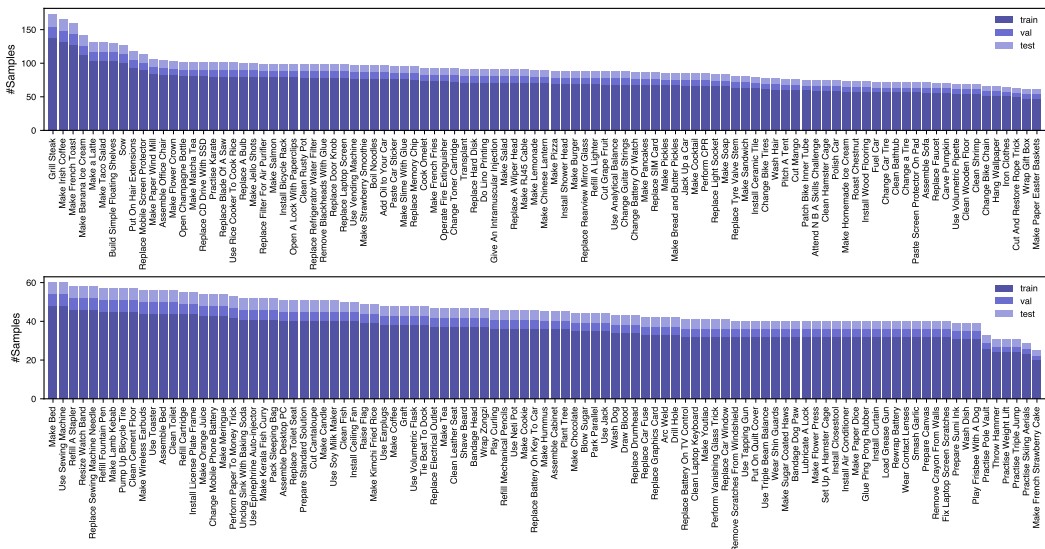

Figure 7: The sample distribution of all the topics in VTT.

Table 6: Statistics of the VTT dataset.

|  | Categories | Topics | Samples | States | Transformations | Unique Transformations |
|---|---|---|---|---|---|---|
| CrossTask | 4 | 18 | 1825 | 12860 | 11035 | 105 |
| COIN | 12 | 180 | 11722 | 56169 | 44447 | 749 |
| Train | 12 | 198 | 10759 | 54716 | 43957 | 853 |
| Val | 12 | 198 | 1352 | 6974 | 5622 | 812 |
| Test | 12 | 198 | 1436 | 7339 | 5903 | 806 |
| Total | 12 | 198 | 13547 | 69029 | 55482 | 853 |

**Extract State Images.** When building the VTT dataset from videos, one important question is how to select images from video frames to construct states. Ideally, the starting state should be an image of the moment just before the transformation starts, and the ending state should be an image of the moment when the transformation ends. Based on CrossTask's precise temporal segment annotations and COIN's precise boundaries of segments annotated with three rounds of refinement, we choose to directly use the start or end frame of the segments as the start or end states. As shown in Figure 1, frames in the boundary of transformations are selected as states. After extracting, we randomly sample 200 examples to check the quality of states and only find a few flaws such as black screens or text transitions. This is mainly caused by careless annotators. As a whole, transformations are predictable from extracted states for humans but sometimes can be challenging. For those difficult samples, it is not enough to reason the transformations by only considering the nearby before and after image states. Our motivation for the masked transformation model is from this observation that transformation should be reasoned in the overall context. Another problem someone may care about is whether there exist multiple transformations between two adjacent states. Since the annotation target of CrossTask and COIN is to segment all key steps in a video, the probability of multiple transformations between states is small.

**Split Dataset.** Finally, we split the data randomly into Train / Val / Test sets with samples of 10759 / 1352 / 1436 in the level of topic. The detailed topic distribution is shown in Figure 7. From the figure, we can see that about half of the topics have over 100 samples. We also summarize the main statistics of the VTT dataset in Table 6.

**Discussions.** Currently, the size of the VTT dataset is small compared with large video datasets such as HowTo100M (Miech et al., 2019), which limits the range of transformations covered by the dataset. The biggest restriction is the high cost of labeling steps/transformations with descrip-

Table 7: The VTT human evaluation guidelines.

| Metric | Score | Criteria |
|---|---|---|
| Fluency | 5 | All sentences are fluent. |
| | 4 | Most sentences are fluent with only a few flaws. |
| | 3 | About half of the sentences are fluent. |
| | 2 | Most of the sentences are difficult to read, only a few are okay. |
| | 1 | All sentences are hard to read. |
| Relevance | 5 | The descriptions are all related to the corresponding before and after images. |
| | 4 | A few descriptions are slightly irrelevant, e.g. the description is related to the underlying topic but cannot be clearly inferred from the images. |
| | 3 | Many descriptions are slightly irrelevant or a few descriptions are irrelevant, e.g. the action or target object mentioned in the transformation does not match the images. |
| | 2 | Many descriptions are irrelevant. |
| | 1 | Most descriptions are irrelevant, or some descriptions are completely irrelevant, e.g. transformation is unrelated to the underlying topic of the images. |
| Logical Soundness | 5 | The underlying logic of the descriptions is consistent with common sense. |
| | 4 | The overall logic is consistent with common sense, with minor flaws. |
| | 3 | There are a few obvious logical problems between the descriptions, e.g. unresonable repeating transformations. |
| | 2 | There are some obvious logical problems, e.g. the order of transformations is obviously not in line with common sense. |
| | 1 | Logic cannot be judged because of the extremely poor fluency or poor relevance leading to overall logic inconsistent with the underlying topic. |

tions and temporal boundaries. One possible way to reduce this cost is first using pretrained step localization models (Wang et al., 2021; Zhang et al., 2022) or action and object state-recognition models (Soucek et al., 2022) to propose coarse steps/transformations and then refine the results with human annotators. With more data to cover large transformations, machines can learn more powerful transformation reasoning models, which have potential enhancement value for many tasks, such as procedure planning, visual storytelling, explaining the phenomenon in life with events, etc. Another issue is the precision of the boundary of existing step segments in CrossTask and COIN. For future construction of larger datasets, we suggest a strategy for possible refinement by applying object state-recognition models (Soucek et al., 2022).

## B    EVALUATION DETAILS

### B.1    AUTOMATIC EVALUATION

We introduce some details that are not included in the main content of the paper when computing automatic metrics. Firstly, we follow the smooth strategy introduced by Chen & Cherry (2014) when computing BLEU@4 to provide more accurate results. This is because descriptions in VTT are usually short, the original BLEU@4 gives a zero score for short texts. In addition, BERT-Score is rescaled with the pre-computed baseline (Zhang et al., 2020) to have a more meaningful score range. BLEU@4 is computed using NLTK package [1]. CIDEr, METEOR, ROUGE, and SPICE are computed with the code from coco-caption [2]. BERT-Score is computed by using the official code [3] provided by the authors.

### B.2    HUMAN EVALUATION

Automatic evaluation metrics have limitations on reflecting the quality of generated text mainly because they are uninterpretable and do not correlate with human evaluations (van der Lee et al., 2019). In the VTT task, we consider three levels of text quality, evaluated by people. The first

---

[1] https://www.nltk.org/api/nltk.translate.bleu_score.html
[2] https://github.com/tylin/coco-caption
[3] https://github.com/Tiiiger/bert_score

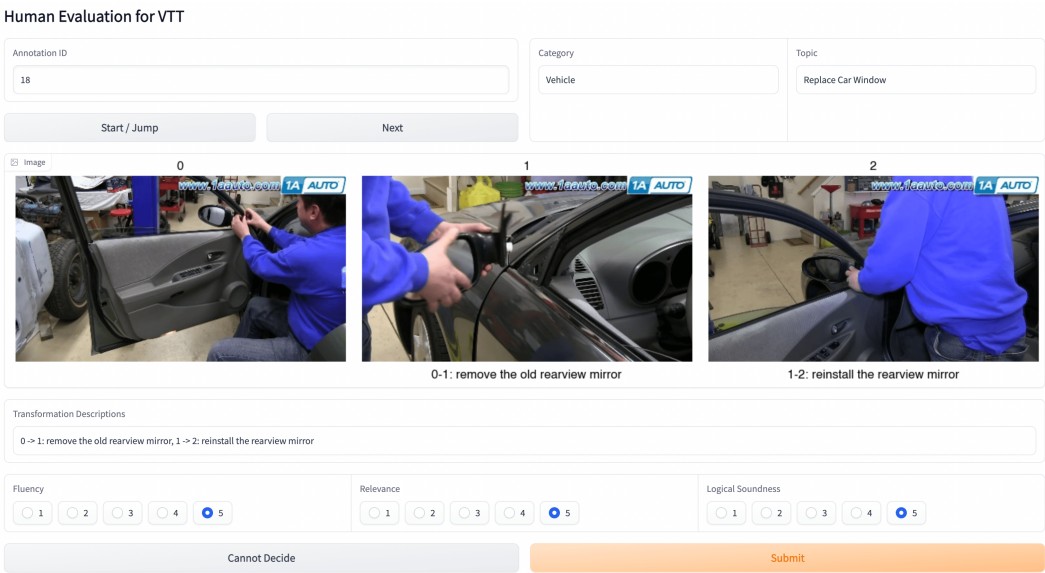

Figure 8: The web interface of human evaluation on VTT.

Table 8: Implementations details of baseline models and TTNet.

| Model | Image Encoder | Context Encoder | Transformation Decoder | Params |
|---|---|---|---|---|
| CST | InceptionV3 | LSTM | LSTM | 379M |
| CST* | CLIP (ViT-L/14) | LSTM | LSTM | 661M |
| GLACNet | ResNet152 | bi-LSTM | LSTM | 128M |
| GLACNet* | CLIP (ViT-L/14) | bi-LSTM | LSTM | 373M |
| DenseCap* | CLIP (ViT-L/14) | Attention | LSTM | 361M |
| TTNet$_{Base}$ | CLIP (ViT-L/14) | Transformer | Transformer | 368M |
| TTNet | CLIP (ViT-L/14) | Transformer | Transformer | 368M |

level considers only the fluency of the text itself. The second level considers the relevance of each individual transformation description to the topic and to the images before and after. The third level considers the logical consistency between transformation descriptions. The assessment uses the 5-point Likert scale and follows the guidelines below in Table 7.

We asked 25 people to evaluate the major baseline models' results shown in Table 1. A subset of samples is selected for human evaluation by randomly sampling testing samples, 1 sample from each topic and 2 extra samples, resulting in 200 samples in total. All annotators are asked to read and follow the guidelines to give their scores. During human evaluation, the annotators can see the images in addition to the category and the topic as a reference. The web interface for human evaluation is shown in Figure 8. Each sample result from each model is evaluated by at least 2 people. Our code of the evaluation webpage will also be released along with the VTT source code.

## C   IMPLEMENTATION DETAILS

During training, we apply commonly used image augmentation tricks, including randomly cropping images into $224 \times 224$ patches, and random flipping. The overall architectures of all baseline models are shown in Table 8. We re-implemented CST and GLACNet following the original paper and their released source code [4] [5]. We didn't find the code of DenseCap and followed their paper to implement the final model. The image encoder of DenseCap is replaced with CLIP since the original model targets video descriptions and uses video encoders. When implementing TTNet, in the

---

[4] https://github.com/dianaglzrico/neural-visual-storyteller
[5] https://github.com/tkim-snu/GLACNet

Table 9: Results of different image encoders.

| | Image Encoder | Params | Acc | B@4 | C | BS |
|---|---|---|---|---|---|---|
| ImageNet Pretrained[7] | InceptionV3 (Szegedy et al., 2016) | 23M | 77.44 | 44.88 | 404.85 | 61.75 |
| | ResNet152 (He et al., 2016) | 59M | 82.82 | 50.71 | 464.01 | 67.40 |
| | ViT-L (Dosovitskiy et al., 2022) | 304M | 85.84 | 58.26 | 540.46 | 73.59 |
| | Swin-L (Liu et al., 2021) | 196M | 86.32 | 57.36 | 531.51 | 73.03 |
| | BEiT-L (Bao et al., 2022) | 306M | 87.48 | 41.57 | 370.00 | 58.80 |
| Image-text Pretrained[8] | RN50 | 39M | 73.30 | 53.35 | 491.80 | 69.79 |
| | RN101 | 57M | 75.70 | 53.78 | 495.30 | 70.08 |
| | ViT-B/32 | 88M | 76.10 | 55.21 | 510.08 | 71.27 |
| | ViT-B/16 | 86M | 80.20 | 57.73 | 534.92 | 73.37 |
| | ViT-L/14 | 304M | 83.90 | **61.22** | **570.63** | **76.25** |

image encoder part, the default CLIP image encoder is ViT-L/14. Image encoders in all models are pretrained and fixed during training. Further details about the image encoder and the comparison can be found in Section D.1. Transformer-based context encoder consists of two transformer encoder layers. The implementation of the transformer is based on x-transfomer [6]. Simplified relative positional encoding (Raffel et al., 2020) is applied in all transformer layers. In the transformation decoder part, we directly borrow CLIP's tokenizer and their vocabulary list. Each transformation description is generated separately with a shared two-layer transformer decoder. The idea of adding transformation representations into word embeddings is inspired by GLACNet (Kim et al., 2019) and we empirically found this way improves a lot on language influence compared with using the representation as the start token. Like the context encoder, simplified relative positional encoding is also used in the transformation decoder. We sample text with top-$k$ top-$p$ filtering with $k = 100$ and $p = 0.9$. The dimension of intermediate vectors is uniformed to be 512, including state representations, transformation representations, and word embeddings. In the training loss part, the adjustment factor $\alpha$ for $\mathcal{L}_{\text{category}}$ is set to be 0.025 and $\beta$ for $\mathcal{L}_{\text{topic}}$ is 0.1. The optimizer we used is AdamW (Loshchilov & Hutter, 2022), with the learning rate first warming up to 1e-4 in the first 2000 steps and then gradually decreasing to 0. All models are implemented with PyTorch (Paszke et al., 2019) and trained on one single Tesla A100 80G GPU card with 50 epochs. The code for training and inference will be released publicly.

# D   MODEL OPTIMIZATION

In this section, we show more detailed information about the selection of the image encoders and the hyperparameters of masked transformation model when optimizing our model.

## D.1   SELECTION OF IMAGE ENCODERS

Image encoding quality is the basis for subsequent reasoning and description of the model, and thus greatly affects the overall performance of the model. From Table 1, we can see that the original version of CST and GLACNet, with Inception V3 and ResNet as image encoders accordingly perform worse than CST* and GLACNet*, indicating the choice of image encoder matters. We conduct a more detailed analysis of the image encoder by testing 10 state-of-the-art image encoders, 5 were pretrained on ImageNet and 5 are CLIP models pretrained on large-scale image-text data. In the table, we show their parameter size, ImageNet top-1 accuracy, and performance on the VTT dataset. We can see that when the parameter sizes are similar, models pre-trained on image and text data perform better than that pre-trained only on image data, e.g. ViT-L/14 vs. ViT-L. This is consistent with the existing understanding that CLIP encodes more semantic information. In addition to training data, factors that affect model performance include model size, patch size used

---

[6]https://github.com/lucidrains/x-transformers

[7]Model weights and top-1 accuracy on ImageNet of ImageNet pretrained models are from: https://github.com/rwightman/pytorch-image-models

[8]Pretrained weights of CLIP models are from https://github.com/openai/CLIP and top-1 accuracy on ImageNet is from Table 10 of the original paper.

in vision transformers, and training strategies. For example, CLIP models have more parameters performs better. While the parameter size between ViT-B/16 and ViT-B/32 are similar, ViT-B/16 encodes finer image has smaller patch size resulting in a better image representation. BEiT-L has the highest accuracy on ImageNet but performs the worst among all models. Our explanation is that BEiT has learned enough image pattern information, but there is a defect in the capture of semantic information.

### D.2 HYPERPARAMETERS OF MASKED TRANSFORMATION MODEL

There are two hyperparameters in masked transformation model, i.e. the mask ratio and the sample ratio. The mask ratio is similar with BERT's mask ratio (Devlin et al., 2019), that is, the percentage of state representations and difference features that are replaced with zero. We compare the mask ratio from 0%-30% and find 15% works the best (Table 10), which is in line with BERT's finding. Another hyperparameter is the sample ratio. The motivation is to tackle the inconsistent issue between training and inference, that is, no features are masked during inference. To fill this gap, samples have opportunities to completely bypass the mask strategy during training. The sample ratio is the probability that the sample will accept the mask strategy. From Table 11, 50% probability is the best, better than the strategy of masking all samples used in BERT.

Table 10: Ablation on the mask ratio.

| Mask Ratio | B@4 | C | BS |
|---|---|---|---|
| 0% | 60.38 | 562.83 | 75.72 |
| 5% | 60.93 | 567.92 | 76.11 |
| 10% | 61.02 | 568.71 | 76.13 |
| 15% | **61.22** | **570.63** | 76.25 |
| 20% | 61.07 | 568.99 | 76.21 |
| 25% | 61.16 | 570.18 | **76.35** |
| 30% | 60.72 | 565.43 | 75.94 |

Table 11: Ablation on sample ratio.

| Sample Ratio | B@4 | C | BS |
|---|---|---|---|
| 0% | 60.38 | 562.83 | 75.72 |
| 25% | 60.39 | 562.15 | 75.63 |
| 50% | **61.22** | **570.63** | **76.25** |
| 75% | 60.96 | 567.99 | 76.00 |
| 100% | 60.95 | 568.18 | 76.10 |

## E GENERALIZATION ANALYSIS

There are two levels of generalization that should be considered in visual transformation telling. At the level of one single transformation, the question is whether machines are able to generalize to different language compositions, i.e. new action-target combinations that do not exist in the training set. At the higher level of multiple transformations, the question becomes to be whether machines can generalize to different transformation combinations, such as different numbers and orders of transformations on the same topic. In this section, we discuss these two kinds of generalization problems and see how well our models perform.

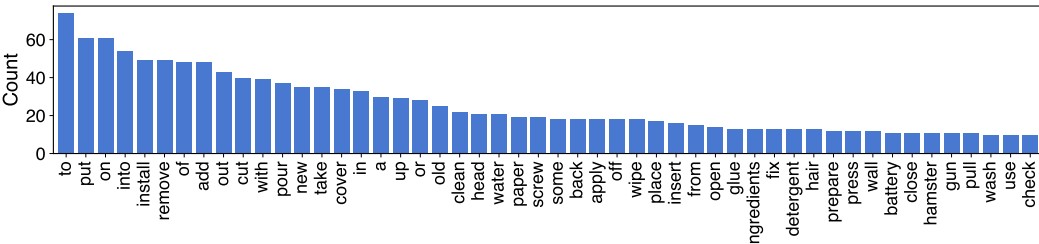

Figure 9: Top 50 words (exclude *the* and *and*) appear multiple times in all 853 unique descriptions.

### E.1 LANGUAGE COMPOSITION

Language composition Johnson et al. (2017) has been widely discussed in the community of visual reasoning that it is very useful if machines are able to generalize to unseen language combinations with seen words. In the VTT dataset, it is common that different transformation descriptions share

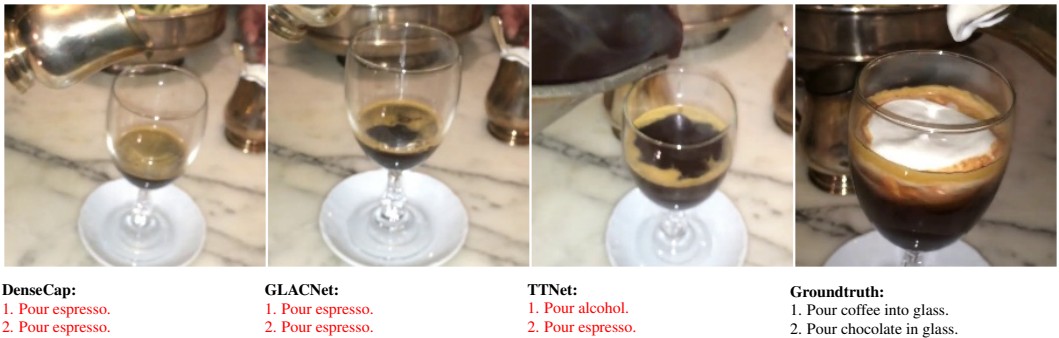

**DenseCap:**
1. Pour espresso.
2. Pour espresso.
3. Add whipped cream.

**GLACNet:**
1. Pour espresso.
2. Pour espresso.
3. Add whipped cream.

**TTNet:**
1. Pour alcohol.
2. Pour espresso.
3. Add whipped cream.

**Groundtruth:**
1. Pour coffee into glass.
2. Pour chocolate in glass.
3. Pour cream.

Figure 10: Models failed to compose unseen transformations with seen words.

Table 12: Statistics of unique transformation combinations. *Val Only* and *Test Only* count combinations that do not exist in the train split.

| Topics | Train | Val | *Val Only* | Test | *Test Only* | All |
|---|---|---|---|---|---|---|
| Dish | 922 | 154 | 95 | 158 | 103 | 1118 |
| Drink and Snack | 804 | 141 | 83 | 146 | 88 | 968 |
| Electrical Appliance | 472 | 99 | 34 | 100 | 41 | 546 |
| Furniture and Decoration | 481 | 83 | 51 | 93 | 45 | 577 |
| Gadgets | 414 | 82 | 33 | 84 | 40 | 487 |
| Housework | 406 | 75 | 48 | 69 | 40 | 494 |
| Leisure and Performance | 408 | 85 | 39 | 85 | 39 | 485 |
| Nursing and Care | 316 | 66 | 29 | 66 | 27 | 371 |
| Pets and Fruit | 208 | 42 | 20 | 45 | 22 | 249 |
| Science and Craft | 395 | 82 | 37 | 87 | 42 | 473 |
| Sport | 167 | 35 | 18 | 38 | 22 | 207 |
| Vehicle | 541 | 110 | 56 | 120 | 50 | 643 |
| Total | 5534 | 1054 | 543 | 1091 | 559 | 6618 |

the same words, especially verbs, and nouns. We count the top 50 words, excluding *the* and *and*, from all 853 unique transformation descriptions in the VTT dataset and results are shown in Figure 9.

With such a large proportion of shared vocabulary, the natural language generation of the transformation is more valuable than the classification of the transformation, since models would have more chances to learn common patterns from transformations with shared words. To investigate whether models learned on the VTT have the generalization ability of language composition, we test models on one manually annotated by us, which comes from a related task in CrossTask, i.e. *Make Bicerin*. The topic contains transformation descriptions that are not included in our training set but are composed without new words. Figure 10 show the results of three models. Unfortunately, all models failed to generate these new descriptions from the existing words but with descriptions that match the states as closely as possible but exist in the list of transformations from the training set. We believe there are two main reasons. Firstly, the small size of unique transformations limits models to gain language composition ability from data. Secondly, current models lack effective design for this generalization ability. Therefore, enlarging the dataset to cover more diverse transformations and designing models with stronger generalization ability will be important future directions for visual transformation telling.

### E.2 TRANSFORMATION COMBINATION

The states and transformations under the same topic can be very different, one important reason is the different combinations of key transformations. For example, *add seasoning* can be the step after the water is boiling, or the noodles are poured, or both, depending entirely on the preference of the person cooking the noodles. This freedom leads to rich transformation combinations even with few key single transformations under each topic. We count the unique combinations in the VTT dataset

Table 13: Performance on all test samples and subsets of test samples whose transformation combinations are shared and non-shared with the training set.

| Model | All | | | | Share | | | | Non-share | | | |
|---|---|---|---|---|---|---|---|---|---|---|---|---|
| | C | Fl | Re | LS | C | Fl | Re | LS | C | Fl | Re | LS |
| CST* | 0.85 | 2.04 | 3.16 | 2.96 | 0.99 | 1.95 | 3.22 | 3.00 | 0.73 | 2.17 | 3.08 | 2.91 |
| GLACNet* | 5.08 | 4.75 | 3.82 | 3.78 | 6.21 | 4.80 | 3.90 | 3.91 | 4.11 | 4.69 | 3.70 | 3.59 |
| DenseCap* | 4.40 | 4.74 | 3.67 | 3.59 | 5.16 | 4.72 | 3.66 | 3.61 | 3.75 | 4.76 | 3.68 | 3.57 |
| TTNet$_{Base}$ | 5.15 | **4.79** | 4.04 | 3.95 | 6.02 | 4.80 | 4.08 | 4.00 | 4.40 | **4.77** | **3.99** | **3.88** |
| TTNet | **5.71** | 4.78 | **4.10** | **4.11** | **7.01** | **4.81** | **4.23** | **4.29** | **4.59** | 4.74 | 3.93 | 3.86 |

as shown in Table 12. From the table, we can see that more than half combinations in the test set do not exist in the training set (559 test only v.s. 1091 total), which means models need a strong generalization ability to predict these unseen transformation combinations well.

To investigate how models perform on these non-share combinations, we separately computed the performance of each model on three subsets of test samples, including all test samples, test samples that have shared combinations with the training set, and test samples that have combinations not included in the training set. From Table 13, we can see that models perform much worse on non-share combinations than shared combinations, e.g. the logical soundness of TTNet is decreased from 4.29 to 3.86. This significant performance degradation suggests that the combination generalization is challenging for existing models. Furthermore, TTNet performs indeed much better on shared combinations than TTNet$_{Base}$ (logical soundness is 4.29 v.s. 4.00), but the situation is not true for non-shared combinations (logical soundness is 3.86 v.s. 3.88), which means the proposed three strategies contribute on learning existing combinations on the training set but has little or even negative effects on model's combination generalization ability.

# F   MORE QUALITATIVE RESULTS.

We show more results in Figure 11 and Figure 12. Figure 12 shows several hard cases that TTNet fails to reason and describe. We point out three potential directions to improve the TTNet. The first one relates the image recognition ability. From the first case, TTNet recognizes the tent as platfond which is wrong. This error might respond to the image encoder that fails to distinguish these objects. Therefore, it may lead to a better result by using a more powerful image encoder. The major point here is that the model needs to identify subtle differences between states and determine specific transformations based on context. The last case is out-of-domain cases, that is, the test samples are quite different from the training data.

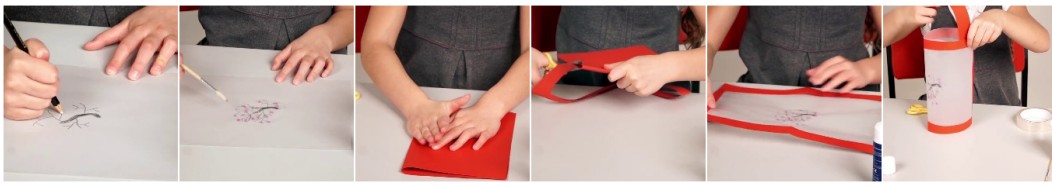

**DenseCap:**
1. Draw black patterns.
2. Cut the raw materials.
3. Cut the packing paper.
4. Cut the raw materials.
5. Nail down or stick together.

**GLACNet:**
1. Fold or bent the paper.
2. Fold or bent the paper.
3. Fold or bent the paper.
4. Nail down or stick together.
5. Nail down or stick together.

**TTNet:**
1. Draw pictures on the materials.
2. Fold or bent the paper.
3. Cut the raw materials.
4. Nail down or stick together.
5. Nail down or stick together.

**Groundtruth:**
1. Draw pictures on the materials.
2. Fold or bent the paper.
3. Cut the raw materials.
4. Nail down or stick together.
5. Nail down or stick together.

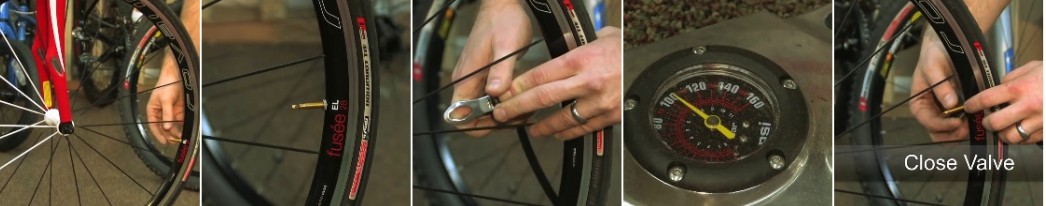

**DenseCap:**
1. Remove the air nozzle.
2. Remove the air nozzle.
3. Pump up to the tire.
4. Pump up to the tire.

**GLACNet:**
1. Screw off the valve cap and open the valve.
2. Pump up to the tire.
3. Pump up to the tire.
4. Tighten the valve and screw on the valve cap.

**TTNet:**
1. Screw off the valve cap and open the valve.
2. Install the air nozzle.
3. Pump up to the tire.
4. Remove the air nozzle.

**Groundtruth:**
1. Screw off the valve cap and open the valve.
2. Install the air nozzle.
3. Pump up to the tire.
4. Remove the air nozzle.

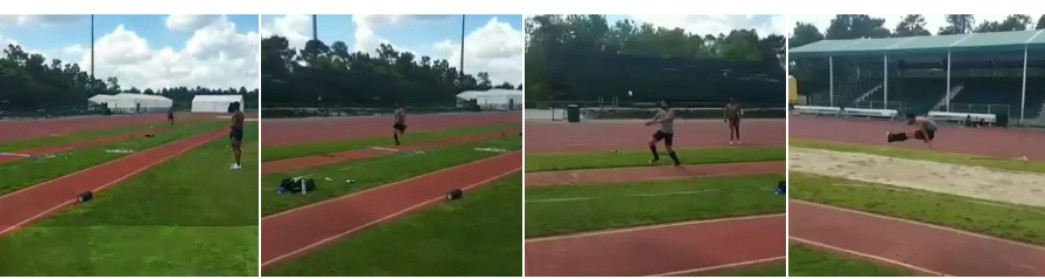

**DenseCap:**
1. Do the first two jumps.
2. Do the first two jumps.
3. Do the third jump.

**GLACNet:**
1. Begin to run up.
2. Do the third jump.
3. Do the third jump.

**TTNet:**
1. Begin to run up.
2. Do the first two jumps.
3. Do the third jump.

**Groundtruth:**
1. Begin to run up.
2. Do the first two jumps.
3. Do the third jump.

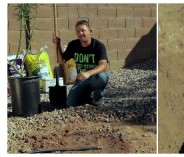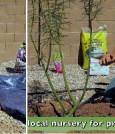

**DenseCap:**
1. Dig a pit with proper size.
2. Fill the tree into the pit.
3. Fill the tree into the pit.
4. Fill the tree into the pit.
5. Fill the tree into the pit.
6. Fill the pit with some soil.

**GLACNet:**
1. Dig a pit with proper size.
2. Fill the tree into the pit.
3. Fill the tree into the pit.
4. Fill the pit with some soil.
5. Fill the pit with some soil.
6. Pour water to the tree.

**TTNet:**
1. Dig a pit with proper size.
2. Fill the tree into the pit.
3. Fill the pit with some soil.
4. Pour water to the tree.
5. Fill the pit with some soil.
6. Pour water to the tree.

**Groundtruth:**
1. Dig a pit with proper size.
2. Fill the tree into the pit.
3. Fill the pit with some soil.
4. Pour water to the tree.
5. Fill the tree into the pit.
6. Pour water to the tree.

Figure 11: More qualitative results in the VTT test data.

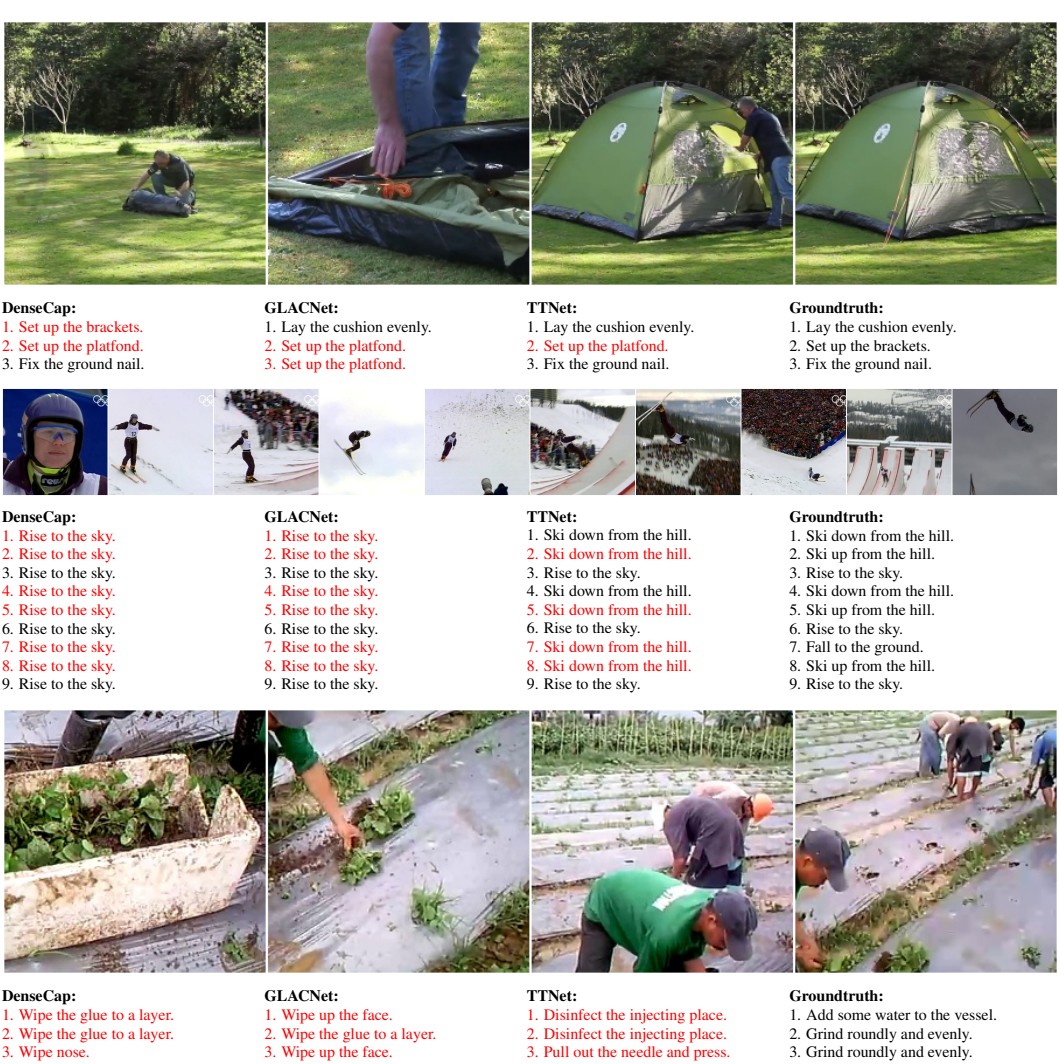

**DenseCap:**
1. Set up the brackets.
2. Set up the platfond.
3. Fix the ground nail.

**GLACNet:**
1. Lay the cushion evenly.
2. Set up the platfond.
3. Set up the platfond.

**TTNet:**
1. Lay the cushion evenly.
2. Set up the platfond.
3. Fix the ground nail.

**Groundtruth:**
1. Lay the cushion evenly.
2. Set up the brackets.
3. Fix the ground nail.

**DenseCap:**
1. Rise to the sky.
2. Rise to the sky.
3. Rise to the sky.
4. Rise to the sky.
5. Rise to the sky.
6. Rise to the sky.
7. Rise to the sky.
8. Rise to the sky.
9. Rise to the sky.

**GLACNet:**
1. Rise to the sky.
2. Rise to the sky.
3. Rise to the sky.
4. Rise to the sky.
5. Rise to the sky.
6. Rise to the sky.
7. Rise to the sky.
8. Rise to the sky.
9. Rise to the sky.

**TTNet:**
1. Ski down from the hill.
2. Ski down from the hill.
3. Rise to the sky.
4. Ski down from the hill.
5. Ski down from the hill.
6. Rise to the sky.
7. Ski down from the hill.
8. Ski down from the hill.
9. Rise to the sky.

**Groundtruth:**
1. Ski down from the hill.
2. Ski up from the hill.
3. Rise to the sky.
4. Ski down from the hill.
5. Ski up from the hill.
6. Rise to the sky.
7. Fall to the ground.
8. Ski up from the hill.
9. Rise to the sky.

**DenseCap:**
1. Wipe the glue to a layer.
2. Wipe the glue to a layer.
3. Wipe nose.

**GLACNet:**
1. Wipe up the face.
2. Wipe the glue to a layer.
3. Wipe up the face.

**TTNet:**
1. Disinfect the injecting place.
2. Disinfect the injecting place.
3. Pull out the needle and press.

**Groundtruth:**
1. Add some water to the vessel.
2. Grind roundly and evenly.
3. Grind roundly and evenly.

Figure 12: Bad cases on the VTT dataset.

