# OpenReview forum: "Visual Transformation Telling"
_ICLR.cc/2023/Conference — Submitted to ICLR 2023_

### Official Review · Reviewer_tWf2 · 2022-10-23

**Confidence:** 3
**Correctness:** 3
**Technical Novelty And Significance:** 3
**Empirical Novelty And Significance:** 3
**Recommendation:** 6

**Clarity, Quality, Novelty And Reproducibility:**

- Clarity: the paper is well-written.
- Quality: the quality of the paper is good.
- Novelty: the authors may need to justify the novelty as well as the application-wise impact of the proposed task.
- Reproducibility: the proposed model and strategies in this paper should be easy to reproduce. In addition, the authors have provided their code in Supplementary Materials but I did not really check.


**Strength And Weaknesses:**

Strength:
1. Experiments and analysis are quite thorough with ablation studies clearly demonstrating the impact of each proposed component of the paper.
2. The proposed model TTNet significantly outperforms the baselines and the proposed strategies are intuitive, reasonable and effective.
3. Interesting qualitative results were presented.
4. Failure cases were shown and discussed.
5. The paper is well-written and easy to follow.

Weakness:
1. The necessity or importance of the proposed new VTT task is not clear. The concerns are:
-  There are similar tasks such as Procedure Planning [1], Walkthrough Planning [1], and Assistant Writer [2] which is an abstract visual storytelling task.
    * Comparison between the VTT task and these tasks should be discussed.
    * The proposed VTT task seems to be a simplified version of the Procedure Planning task. Procedure Planning requires a model to predict the intermediate steps given only the start and end state of a *video*, whereas the VTT task sort of asks a model to predict the step given the start and end state of each *step*. Therefore, Procedure Planning seems to be more complex than VTT. In addition, the application-wise impact of the Procedure Planning task is quite obvious - an agent that is learned to perform Procedure Planning basically learned how to achieve a certain goal/task through some ordered steps. The application-wise importance of the VTT task is currently not clear to me.
2. Whether the current way of constructing the VTT benchmark is sensible is not clear. The reasons are:
    * According to the paper, the first frame of the first transformation (i.e., step) is used as the first state and the last frame of all transformations are used as the remaining states of the video. However, can the start frame and the end frame really display the start state and end state of a step/transformation? Based on my experience with the CrossTask and COIN datasets, very often that the first frame and the end frame of a step cannot truly show the start state and end state of a step. For example, unlike what’s shown in Figure 8, the last frame of the step “Dig a pit with proper size” could be *not* actually displaying the digged pit - the outcome of the transformation, but rather to be a frame just showing the narrator, some tool, etc. There are existing works like "Look for the Change" [3] that aims to find out which 3 frames in the video correspond to the time depicting the object initial state, the state-modifying action and the object end state. Using the method in [3] or designing a method to identify the frames that truly showcase the start and end state of a step, in my opinion, is more sensible.  Otherwise, I’m concerned about how clean the start and end states are in the current VTT benchmark, and how this kind of noise would affect the benchmark.
    * Does the current VTT benchmark really require sophisticated generalizability and reasoning skills from a model? How challenging is the benchmark really? Right now the train/val/test splits of the benchmark were randomly splitted. It is not really clear whether there are any types of biases in the current benchmark, and whether a model can leverage that potential bias to bypass reasoning skills (this is especially important because the authors claimed the proposed benchmark is a reasoning benchmark). Also because the dataset splits were randomly generated, it is not clear to me if the natural language generation of the transformation is necessary, or how important it is for the TTNet to have the natural language generation capability. Since the TTNEt has the natural language generation capability, this means that TTNet should be able to generate unseen verb-noun combinations, unseen order of steps for seen or unseen tasks. After checking the qualitative results, I’m curious about whether the ordered list of transformation descriptions appears in training and whether it is possible that the model simply learns to memorize these descriptions. If the set of transformation descriptions is the same in training and testing, then maybe there is no need to define the VTT task to output the natural language of the transformations; using the step classes like the traditional approaches would be sufficient.
    * One possible improvement to the current VTT benchmark might be providing annotations for a small fraction of the videos from the “Related Task” of the CrossTask dataset and use these videos as testing videos (I’m assuming only videos from the “Primary Task” of the CrossTask dataset were used to form the VTT benchmark since only those videos were provided temporal step boundary annotations). In this way, the VTT training data will have a step transformation like “pour water”, and a step transformation like “pour coffee into glass” will be absent in training but present in testing. Such a testing set would allow one to demonstrate the power of the natural language generation complement.

[1] Chang, Chien-Yi, et al. "Procedure planning in instructional videos." European Conference on Computer Vision. Springer, Cham, 2020.

[2] Ravi, Hareesh, et al. "AESOP: Abstract Encoding of Stories, Objects, and Pictures." Proceedings of the IEEE/CVF International Conference on Computer Vision. 2021.

[3] Souček, Tomáš, et al. "Look for the Change: Learning Object States and State-Modifying Actions from Untrimmed Web Videos." Proceedings of the IEEE/CVF Conference on Computer Vision and Pattern Recognition. 2022.


**Summary Of The Paper:**

Summary:
- The paper proposes a new task called Visual Transformation Telling (VTT), which is defined as predicting the intermediate steps ('steps' are referred to as ‘transformations’ in the paper) in forms of natural language given the start and end state of each step.
- Two existing instructional video understanding datasets, CrossTask and COIN, are combined to form the VTT benchmark.
- A Transformation Telling Net (TTNet), consisting of the CLIP image encoder, a transformer based context encoder and a transformer based decoder for text generation, is proposed to tackle the VTT benchmark problem.
- Three strategies are further proposed to address the challenges of VTT and thus allow the TTNet to be difference sensitive, context aware and context consistent.
- In-depth experiments demonstrate the effectiveness of the proposed model and strategies.


**Summary Of The Review:**

The paper is really well-written, and the performance of the proposed methods is well-supported by experiments. However, the authors may need to further justify the VTT task and how they construct the VTT benchmark.

*After rebuttal*: I appreciate the new analysis added by the authors which to some extent addressed my major concerns on whether the current VTT benchmark really requires generalizability from models and whether the VTT benchmark was reasonably constructed. Therefore I raised the score from 5 to 6. However, there are indeed several aspects of this manuscript that can be improved, e.g., technical novelty of the proposed model - as the authors admitted, the generalizability of the proposed model is not satisfactory. Overall, I think the idea/direction of the paper is worth pursuing, so I encourage the authors to continue improving the paper no matter what the final outcome is.

---

> ### Author Response · Authors · 2022-11-18
> **Response to Reviewer tWf2 (1/2)**
>
> Thank you for the comments and suggestions! Below, we address the comments raised by the reviewer:
>
> **Comparison between the VTT task and similar tasks including Procedure Planning, Walkthrough Planning, and Assistant Writer should be discussed.** Thanks for pointing out these related works. The motivation of procedure planning is to complete a job with given states, while VTT is to explain transformations between states, which has wider scenarios, e.g. explaining the wet ground with rain. Furthermore, the requirement for natural language generation makes VTT have different evaluations and unique challenges such as generalization on language compositions. Walkthrough planning has a different target which is to predict intermediate states. Assistant Writer is a task similar to visual storytelling but stresses more logic among images. The difference is just like visual storytelling, Assistant Writer mixes transformations in the description, making it difficult to evaluate transformation only. We have added these clarifications to the related works.
>
> **VTT task seems to be a simplified version of the Procedure Planning task.** This is not true. Different motivations and different task formats make two tasks different in many directions. Firstly, procedure planning is difficult to evaluate many alternative planning paths because of lacking the environment to execute the plan. VTT provides intermediate states, which reduce the probability of over flexible explanations. In this perspective, the evaluation of VTT is more reliable. Secondly, describing the transformation with language is friendly to scalability and generalization. There is no need to change models when scaling data contains new transformations/actions.
>
> **The application-wise importance of the VTT task is currently not clear to me.** This is a good question. The target of VTT is to test whether machines are able to reason transformations from states like humans. Actually, it is common to have the primary goal of detecting whether a machine has some human ability. Similar tasks include abstract visual reasoning [1], physical reasoning [2], etc. Meanwhile, the ability of reasoning transformations from states is shared in many tasks involving dynamics, including visual storytelling, procedure planning, and video understanding tasks. Addressing the VTT is potentially beneficial to solving these problems.
>
> **Whether the current way of constructing the VTT benchmark is sensible is not clear.** We realize that we missing many details about how we construct the VTT dataset which may lead to doubts about the quality of the dataset. We have added a detailed introduction in Appendix A about how we construct the VTT dataset to make the VTT dataset more transparent and clear. Next, we answer the detailed questions in the following.
>
> **Can the start frame and the end frame really display the start state and end state of a step/transformation?** According to our observations, the answer is yes. CrossTask claims the temporal segment annotations are precise and COIN's boundaries of segments are annotated and refined with three rounds. We do a check by randomly sampling 100 samples in the early stage of dataset construction, and find that the transformation between every two images can be reasoned out with very few exceptions caused by the bad quality of annotation, such as black screens and text transitions.
>
> **There are existing works like "Look for the Change" that aim to find out which 3 frames in the video correspond to the time depicting the object's initial state, the state-modifying action, and the object's end state.** Thanks for suggesting this work. We have read this work and find it may useful for constructing large-scale transformation datasets or improving the quality of states. We have added this work to our discussion in Appendix A about constructing a larger and higher-quality dataset.
>
> **I’m concerned about how clean the start and end states are in the current VTT benchmark, and how this kind of noise would affect the benchmark.** According to our statistics, there are about 1%~2% (about 2 samples in randomly sampled 100 samples) of samples with bad quality annotation, such as black screens and text transitions. Given the annotation is very difficult, we think this is a reasonable and acceptable number. From our results, a few states with bad quality annotation in one example do not affect the overall description quality much, since TTNet is able to cooperate with other states to generate a reasonable description, which is the motivation of MTM.

---

> > ### Author Response · Authors · 2022-11-19
> > **Response to Reviewer tWf2 (2/2)**
> >
> > **Does the current VTT benchmark really require sophisticated generalizability and reasoning skills from a model?** We appreciate this good question. The previous version missed the discussion about the generalization issues on the VTT dataset and we have already this discussion in Appendix E. There are two major generalization problems, i.e. language composition and transformation combination. The **language composition** problem just like you told that *this means that TTNet should be able to generate unseen verb-noun combinations*. We investigate the word frequency of unique transformations in Figure 9 and find that it is very common for different unique transformations to share the same words. However, we test our models on a video from a related task in CrossTask but find it a pity that all models failed to create new transformations with seen words. We also investigate the **transformation combination** problem, which means the same set of transformations can be combined with different numbers and orders. The statistics in Table 12 show that more than half combinations of transformations in the test set do not exist in the training set. We compute the results on different subsets including testing samples that have shared combinations with the training set and have only unique combinations.  The result shows a relatively large decrease in logical soundness from shared to non-shared. Further details can be found in Appendix E.
> >
> > **How challenging is the benchmark really?** Currently VTT is challenging even if only considering the generalization ability of transformation combination. However, we believe there are still many challenges in other directions, such as language composition, and large image diversity. We will try to address these challenges in future work.
> >
> > **Whether the natural language generation of the transformation is necessary.** With such a large proportion of shared vocabulary, the natural language generation of the transformation is more valuable than the classification of the transformation, since models would have more chances to learn common patterns from transformations with shared words. Another consideration is extendability. If we use class labels to represent transformations, adding newly unseen transformations would require changing the model. However, if we use natural language to represent transformations, the model is not needed to change when adding new transformations. We believe this is a more scalable way.
> >
> > **One possible improvement to the current VTT benchmark might be providing annotations for a small fraction of the videos from the “Related Task” of the CrossTask dataset and using these videos as testing videos.** Thanks for the suggestion. We have tested one example in Figure 10 and found that the model is not able to generate new transformations with seen words. We believe there are two main reasons. Firstly, the small size of unique transformations limits models to gain language composition ability from data. Secondly, current models lack effective design for this generalization ability. Therefore, enlarging the dataset to cover more diverse transformations and designing models with stronger generalization ability will be important future directions for visual transformation telling.
> >
> >
> > [1] Zhang, C., Gao, F., Jia, B., Zhu, Y. & Zhu, S. RAVEN: A Dataset for Relational and Analogical Visual REasoNing. in 2019 IEEE/CVF Conference on Computer Vision and Pattern Recognition (CVPR) 5312–5322 (IEEE, 2019). doi:10.1109/CVPR.2019.00546.
> >
> > [2] Bakhtin, A., van der Maaten, L., Johnson, J., Gustafson, L. & Girshick, R. PHYRE: A New Benchmark for Physical Reasoning. in Advances in Neural Information Processing Systems vol. 32 (Curran Associates, Inc., 2019).

---

> ### Author Response · Authors · 2022-12-12
> **Thank you for updating the score**
>
> Dear Reviewer tWf2,
>
> We sincerely thank you for the updated score, and we again appreciate your time and effort in reviewing our paper and reading our comments, which really helped us greatly in improving our paper.
>
> Best,
>
> Authors of  Paper 1398

---

### Official Review · Reviewer_NhJc · 2022-10-23

**Confidence:** 4
**Correctness:** 3
**Technical Novelty And Significance:** 3
**Empirical Novelty And Significance:** 2
**Recommendation:** 5

**Clarity, Quality, Novelty And Reproducibility:**

The paper is generally well-written and clear despite some difficulties in reading (e.g. TVR in abstract without full illustration, Figure 3 could be equipped with better captions and in-figure illustrations, the current one is vague for the autoregressive part though could be understood with text). The proposed Dataset and task are novel but incremental to existing works regarding understanding action-state transitions in content and captioning in task format. The results seem reproducible given the descriptions in the paper.

**Strength And Weaknesses:**

[+] The proposed visual transformation telling task, sharing the same spirit with text-guided visual prediction or dynamics learning, studies important problems including action grounding and world dynamics learning. However, evaluating the description of transformation can be a more direct evaluation compared with image generation metrics since it could be made more structuralized in a quantitative evaluation setting. Therefore I believe, this task could be a good evaluation approach (or checklist) to test models' capabilities on fine-grained world dynamics learning and activity understanding.

[+] The authors provided sufficient ablative studies showing the limitation and strengths of current models and also their proposed TTNet. This could provide new insights into the model design (e.g. masking for autoregressive decoding), incorporating state difference features as context for better transformation learning, etc.

[-] One key concern of this paper also lies in the task design. With the authors evaluating the final transformation as a captioning task, I do not feel that the clarity of text for a direct evaluation has been fully utilized. More specifically, the authors argued that the ultimate goal of the transformation-telling task is to provide a logically consistent explanation for a sequence of observations (i.e. images), however, I do not see any evaluation regarding this part. Can the captioning scores reflect the quality of transformation telling generated by the models? or to what extent should the model elaborate to have a good enough transformation description (e.g., should it be a simple action verb? verb+noun phrase? phrase with adverbs, or what?). Therefore, I do feel that the current task does not suffice as a thorough evaluation benchmark for fine-grained action grounding or action-state association understanding, limiting its significance.

[-] Next, I do have questions regarding the data. The authors select start and end images as states and chain all such states into an image sequence to form the input data for transformation telling given an instructional video. However, I do feel that there are concepts that can not be simply represented by a single image (e.g. reflected by temporal patterns); then did the authors check if there exists (or could be visually distinguished) transformation between each two images so that the transformation could be reasoned out? As I believe the current data sources CrowdTask and COIN contain various instructional activities, I do feel that this could potentially be a problem. Following the same though, several end states could be the result of several previous steps and not only a single step (e.g., soup being red not only because of adding tomato but also adding ketchup), the transformation might not necessarily follow a chain structure, causing ambiguous states for visual transformation telling. These all contribute to the concerns of task quality.

[-] Finally, despite the good performance of the proposed TTNet, I feel that its significance is somewhat limited as there is no specific representation bound to the transformation and therefore incapable of being used as a transformation representation for forward dynamics simulation. The authors could potentially connect with text-guided generation to see if this loop from transformation to text and text to transformation could be closed.

**Summary Of The Paper:**

This paper proposes a new visual reasoning task, Visual Transformation Telling (VTT), to evaluate models' understanding on action-state associations. The authors leveraged instructional videos in existing datasets to form image sequences and ask models to predict the corresponding transformation text in a captioning manner. The authors also proposed an end-to-end transformer-based model for solving this task and show compelling performance results with captioning baselines.

**Summary Of The Review:**

This paper proposes an interesting and potentially significant task, visual transformation telling, for vision-language/video-language learning. However, given the current status of the paper, there are several unclear or unjustified aspects of the paper that still need to be clarified by the authors or considered in future revisions (see weakness). The authors might also want to connect with other works (e.g. text-guided video generation, visual dynamics learning) to better address the significance of their model. Therefore, I'm recommending a reject for this paper and hope the authors could improve in future revisions.

---

> ### Author Response · Authors · 2022-11-18
> **Response to Reviewer NhJc**
>
> Thank you for the comments and suggestions! Below, we address the comments raised by the reviewer:
>
> **Do not feel that the clarity of text for a direct evaluation has been fully utilized.** We agree with you. The previous version only considers automatic metrics, which is not enough. We have added human evaluation results to Table 1 including fluency, relevance, and logical soundness. Human evaluation metrics provide more interpretable results than automatic metrics and help us to understand our models better. We design logical soundness to evaluate how well the overall logic of transformation descriptions conforms to common sense, which now regards the logically consistent.
>
> **Concerns about task quality.** We understand your concerns. We have added a detailed introduction in Appendix A about how we construct the VTT dataset to make the VTT dataset more transparent and clear. Furthermore, we provide more statistics about the VTT dataset. In this introduction, we also mention some annotation information from CrossTask and COIN, which is important for understanding our decisions when building VTT, such as how to choose the start and end states.
>
> **There are concepts that can not be simply represented by a single image (e.g. reflected by temporal patterns).** We are not very sure about this. In our understanding, temporal patterns are related to the transformation process, which is the describing target of VTT.
>
> **Did the authors check if there exists (or could be visually distinguished) transformation between every two images so that the transformation could be reasoned out?** Yes. We do a check by randomly sampling 100 samples in the early stage of dataset construction, and find that the transformation between every two images can be reasoned out with very few exceptions caused by the bad quality of annotation, such as black screens and text transitions. However, this reasoning is not easy even for humans. We find reasoning only according to the adjacent states is not enough. We need to consider the whole process of transformation, such as understanding the overall topic first and reason the transformation from nearby transformations. These observations inspire us to design three effective strategies.
>
> **The transformation might not necessarily follow a chain structure.** This is possible, but overall the probability is low according to our observation.
>
> **The authors could potentially connect with text-guided generation to see if this loop from transformation to text and text to transformation could be closed.** Thanks for the suggestion. We realize that current transformations are quite independent, which reduces the connection between states. Actually, We are considering a causal method to solve this problem, which is close to your ideas. We will try to incorporate this idea in future work.
>
> **Some difficulties in reading.** Thanks for your advice. We have revised the paper to make it more clear.
>
> **The authors might also want to connect with other works (e.g. text-guided video generation, visual dynamics learning) to better address the significance of their model.** Thanks for the suggestion. This is a good direction and we will try to connect with these works in the future work.

---

> > ### Comment · Reviewer_NhJc · 2022-12-01
> > **Post-rebuttal response**
> >
> > Thank the authors for the clarifications in the rebuttal. However, with the current response, I still feel that the interesting point of this abductive task has not been thoroughly studied given the complexity of the task and also the evaluation of logical consistency (as a benchmark we can not run human evaluation each time on new generation results for comparison). Therefore, I increase my ratings to a weak reject as the data curation process becomes clear, and hope that the authors could improve the discussion on these topics and connect it with related fields to make a stronger submission.

---

> > > ### Author Response · Authors · 2022-12-06
> > > **Response to Post-rebuttal response (1/2)**
> > >
> > > Thank you again for your response and your comment. We are happy to further explain your concerns to clear your doubts.
> > >
> > > **The evaluation of logical consistency: we can not run human evaluation each time on new generation results for comparison.** We understand it would be better to design an automatic metric for logical consistency, but designing such an automated metric is very challenging. Actually, in the field of text generation, the problem of designing automatic metrics correlated with human evaluation has been studied for decades. The metrics we used, including BLEU@4, METEOR, ROUGE-L, CIDEr, SPICE, BERT-Score, all contribute to this direction. However, these metrics still have problems that are uninterpretable [1] and have been argued that are not suitable to assess linguistic properties [2]. In the first version of this paper, we followed the evaluation protocol in previous works that are most related to ours, including visual storytelling [3], change captioning [4], and visual abductive reasoning [5]. We learned from reviewers' comments on the first version and realized that the evaluation of the logical consistency is missing. Since logical consistency is one of the major challenges of our task, we designed evaluation criteria, completed the human evaluation, and add the results to our paper. From the results, we indeed interpret the results and understand models more clearly.
> > >
> > > To reduce the cost of human evaluation on logical consistency, we will release the code of the human evaluation tool and detailed sample-wise scores from our human evaluation for convenient comparison. Additionally, from Table 1, logical soundness and automatic metrics, such as BERT-Score, are consistent in their overall trends. Therefore, these automatic metrics are valuable as low-cost indicators of logical consistency trends.
> > >
> > > [1] Van Der Lee, C., Gatt, A., Van Miltenburg, E., Wubben, S. and Krahmer, E., 2019. Best practices for the human evaluation of automatically generated text. In Proceedings of the 12th International Conference on Natural Language Generation (pp. 355-368).
> > >
> > > [2] Scott, D. and Moore, J., 2007, April. An NLG evaluation competition? eight reasons to be cautious. In Proceedings of the Workshop on Shared Tasks and Comparative Evaluation in Natural Language Generation (pp. 22-23).
> > >
> > > [3] Huang, T.H., Ferraro, F., Mostafazadeh, N., Misra, I., Agrawal, A., Devlin, J., Girshick, R., He, X., Kohli, P., Batra, D. and Zitnick, C.L., 2016, June. Visual storytelling. In Proceedings of the 2016 conference of the North American chapter of the association for computational linguistics: Human language technologies (pp. 1233-1239).
> > >
> > > [4] Park, D.H., Darrell, T. and Rohrbach, A., 2019. Robust change captioning. In Proceedings of the IEEE/CVF International Conference on Computer Vision (pp. 4624-4633).
> > >
> > > [5] Liang, C., Wang, W., Zhou, T. and Yang, Y., 2022. Visual Abductive Reasoning. In Proceedings of the IEEE/CVF Conference on Computer Vision and Pattern Recognition (pp. 15565-15575).

---

> > > > ### Author Response · Authors · 2022-12-06
> > > > **Response to Post-rebuttal response (2/2)**
> > > >
> > > >
> > > > **Connection with related fields.** We would like to discuss more on the connection with related fields. Actually, Reviewer tWf2 asked two similar questions in the review comments, including the application of VTT and the relation with Procedure Planning, Walkthrough Planning, and Assistant Writer. We expect these concerns have been addressed since Reviewer tWf2 has increased their score but without a response comment. We add some discussions here and borrow some content from the answers to these questions which have been already added to our paper.
> > > >
> > > > - **The application-wise importance of the VTT task?** The target of VTT is to test whether machines are able to reason transformations from states like humans. Actually, it is common to have the primary goal of detecting whether a machine has some human ability. Similar tasks include abstract visual reasoning [1], physical reasoning [2], etc. Meanwhile, the ability of reasoning transformations from states is shared in many tasks involving dynamics, including visual storytelling, procedure planning, and video understanding tasks. Addressing the VTT is potentially beneficial to solving these problems.
> > > >
> > > > - **Comparison between the VTT task and similar tasks including Procedure Planning, Walkthrough Planning, and Assistant Writer.** The motivation of procedure planning is to complete a job with given states, while VTT is to explain transformations between states, which has wider scenarios, e.g. explaining the wet ground with rain. Furthermore, the requirement for natural language generation makes VTT have different evaluations and unique challenges such as generalization on language compositions. Walkthrough planning has a different target which is to predict intermediate states. Assistant Writer is a task similar to visual storytelling but stresses more logic among images. The difference is just like visual storytelling, Assistant Writer mixes transformations in the description, making it difficult to evaluate transformation only. We have added these clarifications to the related works.
> > > >
> > > > - **Relation to visual dynamics learning, text-guided video generation.** Visual dynamic learning is highly related to VTT. The motivation of our task is to let machines learn and tell the dynamics between states, and provide a test bed to evaluate this ability. Text-guided video generation has been very popular due to recent progress in diffusion models. This problem can be regarded as a reverse problem of our task, i.e. given text and generate frames. This is a promising idea if we are able to control video generation through some key transformation descriptions. As far as we know, there is no existing work is based on this idea. We are interested to explore this direction in the future.
> > > >
> > > > [1] Zhang, C., Gao, F., Jia, B., Zhu, Y. & Zhu, S. RAVEN: A Dataset for Relational and Analogical Visual REasoNing. in 2019 IEEE/CVF Conference on Computer Vision and Pattern Recognition (CVPR) 5312–5322 (IEEE, 2019). doi:10.1109/CVPR.2019.00546.
> > > >
> > > > [2] Bakhtin, A., van der Maaten, L., Johnson, J., Gustafson, L. & Girshick, R. PHYRE: A New Benchmark for Physical Reasoning. in Advances in Neural Information Processing Systems vol. 32 (Curran Associates, Inc., 2019).

---

> > > ### Author Response · Authors · 2022-12-12
> > > **Have your concerns been addressed?**
> > >
> > > Dear Reviewer NhJc,
> > >
> > > Thank you a lot for your constructive and helpful comments and the updated score. We have replied to your post-rebuttal response. Could you help to check if our reply has addressed your concerns? Thank you!
> > >
> > > Best Regards,
> > >
> > > Authors of Paper 1398

---

### Official Review · Reviewer_ipHd · 2022-10-25

**Confidence:** 4
**Correctness:** 3
**Technical Novelty And Significance:** 2
**Empirical Novelty And Significance:** 3
**Recommendation:** 5

**Clarity, Quality, Novelty And Reproducibility:**

The writing of this paper is clear, and their main contribution is the data and the proposed method. The model contribution of this paper is limited, and some data analysis is not solid also. Some implementation details are not described clearly in this paper.

**Strength And Weaknesses:**

The difference in state representations or difference between visual features has been widely used in video understanding tasks. The difference in sensitive encoding in this paper is not new since lots of video captioning also has a close idea.

The masking mechanism in this paper is not to predict the specific ground truth, such as words or attributes; instead, they mask the input features and predict the representation. However, this paper lacks details, and the authors should describe more about the masking and loss function design.


Also, since the feature keeps changing, masking in the feature space makes the MTM harder. Instead of masking 15%, did the authors try other masking ratios?

In their setting, they miss the SPICE score, which is also widely used in the image captioning domain. Also, considering the gap between machine evaluation and human judgment, it is necessary to have humans evaluate the predictions captions. Since the visual description in this paper is much longer than image captioning, simply replying to language rules may not well reflect the real performance of predictions.

CLIP model was pretrained on image-text data, while the data in this paper is video data; considering the gap between images and videos, what are the effects of the pretrained model for this setting? Did the authors consider some video-text pretrained model for this task?

**Summary Of The Paper:**

This paper proposes a new task as well as a dataset that was collected from instructional video datasets. Apart from this, they also have a model proposed for this task. They analyze the proposed data and achieve good performance with their model.

**Summary Of The Review:**

Generating descriptions for video has been a long time, and this paper proposes a new task called Visual Transformation Telling. This task is new and interesting, and the baseline models reported in this paper cover most of the recent frameworks. However, considering the dataset is their major contribution, I still think this paper still has some room to improve.

---

> ### Author Response · Authors · 2022-11-18
> **Response to Reviewer ipHd**
>
> Thank you for the comments and suggestions! Below, we address the comments raised by the reviewer:
>
> **Difference between visual features has been widely used in video understanding tasks.** That's true. I may need first to clarify the main difference between VTT and a video captioning task. Video provides the full process of transformation throughout dense frames, reducing the need for reasoning. In contrast, the intermediate information, namely transformation, is missing in VTT, which is just the target we want models to describe. To reason the missing transformation, it is natural to start from the differences between the adjacent states because they are caused by transformations. Results in Table 3 show these simple features work very well. Since VTT is a new task, we want to deliver insights like this to people with some simple and effective methods.
>
> **This paper lacks details, and the authors should describe more about the masking and loss function design.** Sorry for the lack of details. Your understanding of masking the input features and predicting the transformation representations is right. The mask strategy is just like the masked language model in BERT and the loss function is a combination of text generation loss $\mathcal{L}_\text{text}$ (eq.2) and two weighted cross entropy losses including $\mathcal{L}_\text{category}$ and $\mathcal{L}_\text{topic}$. We have added more details in the revised version.
>
> **The feature keeps changing, masking in the feature space makes the MTM harder.** The image encoder is fixed during training, therefore, the state features and difference features are not changing sharply. In practice, the whole training process is very stable.
>
> **Did the authors try other masking ratios?** Yes. we have added the results to Table 10 in Appendix D.2. The conclusion is that 15% works the best, which is in line with the results in BERT.
>
> **Miss the SPICE score.** Thanks for the suggestion. We have added the SPICE score to Table 1.
>
> **It is necessary to have humans evaluate the predictions captions.** Very good suggestion. We have asked 25 people to complete the human evaluation. Results in Table 1 include fluency, relevance, and logical soundness. Human evaluation metrics provide more interpretable results than automatic metrics and help us to understand our models better. We also describe the detailed guideline of humane evaluation in Appendix B.2.
>
> **What are the effects of the pretrained model?** In our experiments, without pretrained image encoders, the whole model is hard to converge. We compare a lot of different pretrained image encoders in Table 9. The best model is ViT-L/14 from CLIP, better than the best model pretrained on ImageNet, which suggests vision-and-language pretraining is more effective than vision-only pretraining for VTT.
>
> **Did the authors consider some video-text pretrained model for this task?** Not yet. The input of VTT is a sequence of images corresponding to the states. This is different from the input of video-text tasks. However, the video-text pretraining model may imply some features related to dynamic changes. We would like to explore this in future work.
>
> **Some data analysis is not solid.** We newly add Appendix A to provide a detailed introduction about how we construct the VTT dataset and provide more statistics about the VTT dataset.
>
> **Some implementation details are not described clearly.** In Appendix C, we complete the missing implementation details in the revised version.

---

> ### Author Response · Authors · 2022-12-07
> **Have all your concerns been addressed?**
>
> Dear Reviewer,
>
> We would like to gently remind you that the discussion period is ending soon. We have answered every question from your review comments and modified our paper according to your comment, including:
> - adding human evaluation,
> - adding the detailed introduction of the VTT dataset,
> - adding missing implementation details,
> - adding ablation studies on mask ratio and sample ratio of MTM,
> - adding the SPICE score in Table 1,
> - adding some details about MTM and loss design in Section 4.
>
> We kindly ask whether you still have some questions and we are happy to answer them. If not, would you please raise your rating accordingly? Thank you for your time and consideration.

---

> ### Author Response · Authors · 2022-12-12
> **Your feedback is important to us**
>
> Dear Reviewer ipHd,
>
> Thank you a lot for your constructive and helpful comments. Could you help to check if our reply has addressed your concerns? Thank you!
>
> Best Regards,
>
> Authors of Paper 1398

---

### Author Response · Authors · 2022-11-18
**Reviewer comments incorporated in the newest revision**

Thanks to all the reviewers for the helpful suggestions! We’ve uploaded a new version of the paper that incorporates some comments from reviewers. These include:

- **[R1, R2]** In Table 1, we add human evaluation results including fluency, relevance, and logical soundness. In Appendix B.2, the detailed guideline for humane evaluation is provided.
- **[R1, R2, R3]** In Appendix A, we newly add a detailed introduction about how we construct the VTT dataset and provide more statistics about the VTT dataset. In Section 3.2 we modify a few sentences about the introduction of the VTT dataset to be more clear.
- **[R3]** In related works, we clarify how VTT differs from similar tasks including Procedure Planning, Walkthrough Planning, and Assistant Writer.
- **[R3]** In the newly added Appendix E, we discuss the generalization problem on VTT, including language composition and transformation combination.
- **[R1]** In Appendix C, we add missing implementation details.
- **[R1]** In the newly added Appendix D.2, we add ablation studies on mask ratio and sample ratio of MTM to the appendix. Results are shown in Table 10 and Table 11.
- **[R1]** In Table, we add the SPICE score.
- **[R2]** In Figure 3, we add descriptive captions and refine the figure to be more clear.
- **[R1]** In Section 4, we add some details about MTM and loss design.

---

### Author Response · Authors · 2022-12-08
**Response to Reviewer Comments**

As the discussion period is nearing its end, we would appreciate it if the reviewers could take the time to read our replies and respond to them. We would be glad to answer any concerns remaining after our revision. Thank you for your time and consideration.

---

### Decision · Program_Chairs · 2023-01-20

**Decision:**

Reject

**Justification For Why Not Higher Score:**

Ultimately, the overall recommendation to reject is centred around two key aspects.

1) Interesting premise, but operationalization / implementation of the VTT task raises doubts because several key aspects of the task definition and design did not come across as well thought-out.

2) Instead of choosing to do a "dataset paper" and focusing on the quality and characteristics of the task and data, etc. typical of such papers, this paper had in equal parts a new task and a new model, neither of which were done particularly well.

This is certainly not a bad paper at all, with no obvious fatal flaws, but reviewers gave only lukewarm support (both numerically and in writing). Overall, there seems to be unanimous feeling (among reviewers and myself) that the paper is "not quite there yet" for the various reasons stated.

Personally though, I highly agree with reviewer "tWf2" when they state that "overall, I think the idea/direction of the paper is worth pursuing, so I encourage the authors to continue improving the paper no matter what the final outcome is."


**Justification For Why Not Lower Score:**

n/a

**Metareview: Summary, Strengths And Weaknesses:**

This paper proposes a new visual reasoning task, called Visual Transformation Telling (VTT). For a given sequence of images, the VTT task is to provide natural language descriptions of the transformations between every two neighboring images. Unlike previous related tasks, which focus on just linking the states, VTT focuses on the transformational processes that link the states. The VTT task is operationalized by building on existing instructional video datasets, namely CrossTask and COIN. The paper proposes a new end-to-end learnt model, Transformation Telling Net (TTNet), which consists of an image encoder, context encoder (to link images) and a decoder (to generate text). The base TTNet is enhanced to boost performance through several ways, such as usage of difference features to boost sensitivity to changes. Comparisons with captioning baselines demonstrate good performance by the enhanced TTNet. The evaluations include both automatic evaluations, as well as human evaluations by 25 evaluators looking at fluency, relevance and logical soundness.


-- STRENGTHS --

1) The proposed task is conceptually interesting and novel.

2) The experiments and analyses (including ablations, qualitative results, failure cases) are fairly thorough.

3) The proposed model outperforms the various baselines.


-- WEAKNESSES --

Putting aside relatively minor weaknesses and those that have been adequately addressed or clarified, the remaining key weaknesses are as follows.

1) Evaluation of what really matters. In response to initial reviews raising issues about absent / missing evaluation of the logical consistency of the explanations (a purported key aspect of the VTT Task), the authors managed to add in subjective evaluations by 25 human evaluators. The subjective evaluations included fluency, relevant and logical consistency. However, there are remaining reasonable doubts about the feasibility of VTT due to necessity of using human evaluators -- logical consistency is a key aspect of VTT, and this can only really be evaluated by a panel of humans. While not a fatal issue, the paper would have been much stronger if a reasonable alternative had been developed. The authors did not satisfactorily address this in the rebuttals, ultimately. Even putting aside this issue, their human evaluation of logical consistency is a very coarse-grained, uni-dimensional one; the VTT task is most interesting because it holds the potential to address complex issues involving causality, contextual dependencies, commonsense knowledge, etc., but these are all folded into a single number.

2) While the initial premise of the VTT task is very interesting, the actual definition of it comes across as overly simplified, not well thought through, and with many unstated assumptions that are not very general. Roughly speaking, VTT assumes a few things such as meaningful transformations between steps (but what if they are just adjacent frames in a 120 FPS video, with little meaningful change? or take some extreme opposite), a linear chain of transformations, that the transformations are one-off, that the transformations nicely complete between two adjacent frames, etc. etc. It so happens that VTT works okay for the instructional video datasets chosen (because of their nature), but that severely limits the generality and applicability of the task in practice.

3) Technical novelty of the proposed model is weak, putting together existing components and known "bells and whistles".




**Summary Of Ac-Reviewer Meeting:**

n/a